# PERISHABLE ONLINE INVENTORY CONTROL WITH CONTEXT-AWARE DEMAND DISTRIBUTIONS

## ABSTRACT

We study the online contextual inventory control problem with perishable goods. In this work, we propose and consider a more realistic—and more challenging— setting where both the expected demand and the (residual) noise distribution depend on the observable features. Surprisingly, little is known when the noise is context-dependent, which captures the heteroskedastic uncertainty in demand that is important in inventory control. The optimal inventory quantity in this general setting is no longer a linear function of features (unlike the case when the expected demand is linear and the noise is i.i.d.), making online gradient descent—the gold standard therein—inapplicable. We first propose an algorithm that achieves the near-optimal regret $\widetilde{O}(\sqrt{dT} + T^{\frac{p+1}{p+2}})$ under linear expected demand and context-aware noise. Here $d$ is the feature dimension, and $p \leq d$ is an underlying dimension that captures the intrinsic complexity of the noise distribution. When the expected demand is nonlinear, we propose to use neural networks to capture the nonlinearity, and prove a regret bound $\widetilde{O}(\sqrt{\alpha T} + T^{\frac{p+1}{p+2}})$ under over-parameterized networks, where $\alpha$ depends on the nonlinear demand complexity and the network architecture. Additionally, under mild regularity conditions on the noise, the exponential factor $T^{\frac{p+1}{p+2}}$ in these regret bounds is improved to $p\sqrt{T}$. Finally, we provide a matching minimax lower bound $\Omega(\sqrt{dT} + T^{\frac{p+1}{p+2}})$ under linear expected demand. To our best knowledge, our results provide the first minimax optimal characterization for online inventory control with context-dependent noise and the first theoretical guarantees when the expected demand is nonlinear in features.

## 1 INTRODUCTION

Inventory control under uncertain demand is a central problem in operations management. In many real-world systems, a decision-maker (DM) must repeatedly choose inventory levels over a time horizon, facing random demand and incurring overstocking or understocking costs (Zipkin, 2000). A widely used modeling approach assumes that the demand at time $t$ takes the form $D_t = \boldsymbol{\theta}_*^\top \boldsymbol{x}_t + \epsilon_t$, where $\boldsymbol{\theta}_* \in \mathbb{R}^d$ is an unknown parameter, $\boldsymbol{x}_t \in \mathbb{R}^d$ the observable context variables, and $\epsilon_t$ an i.i.d. random noise independent of context (Ban & Rudin, 2019; Ding et al., 2024; Huang et al., 2025).

However, despite its statistical simplicity and interpretability, this linear model can fail in practice. In many applications, the variability of demand depends strongly on the contextual information. For example, in e-commerce platforms, demand uncertainty can vary with user types, geographic regions, or temporal factors such as holidays or promotions. Such heteroskedasticity is well-documented in the empirical inventory literature (Zhang, 2007; Kanet et al., 2010; Katanyukul et al., 2011), yet this cannot be captured by the standard homoskedastic model.

To gain more insights on when the heteroskedasticity will occur, consider a natural e-commerce setting where customer-level purchase decisions are modeled as independent Bernoulli events: at time $t$, the demand arises from $n$ independent customers, each purchasing with probability $p(\boldsymbol{x}_t)$ depending on the context $\boldsymbol{x}_t$. Then the aggregate demand $D_t$ follows a Binomial distribution with mean $np(\boldsymbol{x}_t)$ and variance $np(\boldsymbol{x}_t)(1 - p(\boldsymbol{x}_t))$, both of which are context-dependent. For instance, if we consider the sales of umbrellas, when there is zero precipitation, $p(\boldsymbol{x}_t)$ is close to 0 and the demand is almost deterministic. When the precipitation level is intermediate, $p(\boldsymbol{x}_t)$ can be at a constant level, leading to an $O(n)$ variance. Nonetheless, this simple example is not captured by the standard

demand model even if $np(\boldsymbol{x}_t)$ is linear in $\boldsymbol{x}_t$, and failing to properly capture the heteroskedasticity can lead to significant additional loss, as highlighted by the empirical studies above.

In this paper, we study online inventory control with a *context-aware* demand distribution, in a general semi-parametric framework. We crucially allow the noise distribution to vary with context through a potentially lower-dimensional feature $\boldsymbol{z}_t \in \mathbb{R}^p$. Here $\boldsymbol{z}_t$ may be either a transformation of $\boldsymbol{x}_t$, a subset of it, or simply $\boldsymbol{x}_t$ itself; and $p \leq d$ reflects the intrinsic complexity of the distributional dependence of noise on the context. Note $p = 1$ in the Binomial model above. Our contributions are threefold:

- We formalize the setting of online contextual inventory control with context-aware demand distributions, and characterize a minimax regret lower bound $\Omega(\sqrt{dT} + T^{\frac{p+1}{p+2}})$ for the linear demand model.
- We propose an algorithm that achieves the near-optimal regret $\widetilde{O}(\sqrt{dT} + T^{\frac{p+1}{p+2}})$ for the linear model and $\widetilde{O}(\sqrt{\alpha T} + T^{\frac{p+1}{p+2}})$ for general nonlinear demand model that the literature has not addressed. $\alpha$ characterizes the complexity of the nonlinear model.
- Under mild regularity conditions on noise, the regret guarantee is improved to $\widetilde{O}(\sqrt{dT} + p\sqrt{T})$ and $\widetilde{O}(\sqrt{\alpha T} + p\sqrt{T})$ respectively.

## 1.1 LITERATURE REVIEW

When the demand distribution is known, the classic newsvendor model provides a closed-form solution for the optimal order quantity (Zipkin, 2000). However, this assumption is rarely satisfied in practice, and the DM must learn the demand structure on the fly. This has led to a surge of interest in the online learning-while-optimizing paradigm in inventory control (Huh & Rusmevichientong, 2009; Chen & Chao, 2020; Zhang et al., 2020; Davoodi et al., 2022). Many of these works build on Online Stochastic Gradient Descent (OSGD), exploiting the convexity of the expected loss function in the inventory level.

These studies, however, often ignore demand-side covariates. More recently, contextual inventory control has received increasing attention, where the demand depends on observable features such as weather, product metadata, or customer information; see Ban & Rudin (2019); Xu et al. (2023); Zhao et al. (2024); Zhang et al. (2024); Qi et al. (2024); Bertsimas & Kallus (2020); Fu et al. (2024). Most of these works focus on offline learning or distributionally robust optimization, and do not address the online decision-making setting. To the best of our knowledge, the only work that directly addresses the online contextual inventory control problem is Ding et al. (2024), which studies the linear demand model with context-independent noises. Their results crucially rely on that the optimal context-dependent solution is linear in the context under i.i.d. noise, allowing them to compute the loss gradient and apply OSGD. In contrast, we study a more general setting where the noise distribution may depend on the context, rendering gradient-based methods unreliable.

To tackle this problem, we take an estimation-to-decision perspective and draw inspiration from the linear bandit literature. Linear bandits address the setting where the DM repeatedly chooses from $K$ actions, each assigned with an action-specific context at every time $t$, and collects rewards linear in the chosen action's context (Li et al., 2010; Zhou et al., 2020). While not directly applicable, this line of research provides rich algorithmic ideas for estimating underlying linear structures under complicated dependencies in online learning. These ideas shed light to various other fields of contextual operations management, including dynamic pricing (Cohen et al., 2020; Tullii et al., 2024) and online advertising (Badanidiyuru et al., 2023; Wen et al., 2025). Our work leverages some of these insights and contributes to this broad class of literature.

**Notations** Denote $(x)^+ := \max\{x, 0\}$ for $x \in \mathbb{R}$ and $[N] = \{1, 2, \ldots, N\}$ for $N \in \mathbb{N}$. Let $\|\boldsymbol{x}\|_{\boldsymbol{A}} := \sqrt{\boldsymbol{x}^\top \boldsymbol{A} \boldsymbol{x}}$ for positive semi-definite matrix $\boldsymbol{A}$. For a cumulative distribution function (CDF) $Q$, we use $Q^{-1}$ to denote the corresponding quantile function. We use the filtration $\mathcal{F}_t$ to denote the available information up to the beginning of time $t$, and write $\mathbb{E}_t[X] = \mathbb{E}[X|\mathcal{F}_t]$ for any $\mathcal{F}_t$-measurable random variable $X$. We use the standard notations $O, \Omega, \Theta$ to denote the asymptotic behaviors. In addition, we use $\widetilde{O}(\cdot)$ to hide logarithmic factors and the dependence on parameters other than $d, T$, and $p$ (i.e., it hides the parameters $b, h, M, L$).

## 2 PROBLEM SETUP

We consider the online inventory control problem with contextual information. Over a time horizon $t \in [T]$, the DM observes a context vector $\boldsymbol{x}_t \in \mathbb{R}^d$ and chooses an inventory level $c_t \in [0, M]$, where $M \geq 0$ is a known upper bound for the demand. Then a random demand $D_t \in [0, M]$ is realized, and the DM incurs a loss (conditioned on the observed context $\boldsymbol{x}_t$) defined as follows:

$$\ell_t(c_t) = h\mathbb{E}\Big[(c_t - D_t)^+ \,\big|\, \boldsymbol{x}_t\Big] + b\mathbb{E}\Big[(D_t - c_t)^+ \,\big|\, \boldsymbol{x}_t\Big],$$

where the first part corresponds to overstocking with a holding cost of $h$ per unit, and the second part corresponds to understocking with a lost-sale opportunity cost of $b$ per unit. At the end of each period $t$, the DM observes demand realization $D_t$, and any amount of overstocking perishes. The DM's objective is to minimize the cumulative loss over $T$ periods.

In terms of modeling, we assume that the demand satisfies $D_t = f_*(\boldsymbol{x}_t) + \epsilon_t$, where $f_* : \mathbb{R}^d \to [0, M]$ is an unknown model characterizing the demand mean, and $\epsilon_t$ is a mean-zero noise. We will cover both linear and nonlinear $f_*$ in Section 3.1. Importantly, we allow the distribution of $\epsilon_t$ to be time-varying: let $Q_t(\cdot) \equiv Q(\cdot; \boldsymbol{z}_t)$ denote the conditional CDF of $\epsilon_t$ given relevant features $\boldsymbol{z}_t \in \mathbb{R}^p$, and the DM only observes $\widehat{\boldsymbol{z}}_t \in \mathbb{R}^p$ which may be different from $\boldsymbol{z}_t$. One can consider $\boldsymbol{z}_t$ as a transformation or subset of the context $\boldsymbol{x}_t$. For clarity, throughout this work, we refer to $\boldsymbol{x}_t$ as *context* and $\boldsymbol{z}_t$ as *feature*. This model is flexible and thereby powerful in the following sense:

**Encoding prior** The dependence of $Q$ on $\boldsymbol{z}_t$ is unrestricted, so the DM has full flexibility in choosing which features to include in $\boldsymbol{z}_t$ and in what form. For instance, if $Q$ depends only on a single temporal variable within the context $\boldsymbol{x}_t$ and the DM knows this, then setting $\boldsymbol{z}_t$ to be this temporal factor with $p = 1$ suffices. Otherwise, the DM may include a larger subset of $\boldsymbol{x}_t$ to capture the potential dependence, which increases the feature dimension $p$. In this way, the DM's prior belief about the structure of $Q$ is encoded directly through the construction of $\boldsymbol{z}_t$ and is reflected in the dimension $p$.

**Learning features** We allow the observed features $\widehat{\boldsymbol{z}}_t$ of the DM to deviate from the true $\boldsymbol{z}_t$, which models the scenario when the DM adopts features that are themselves learned online. Examples include (1) the estimated demand mean and/or (2) predictions generated by black-box oracles such as language models that get updated when new data becomes available.

Throughout this work, we make the following assumption on the CDF:[1]

**Assumption 1** (Lipschitz CDF). *The noise CDF $Q(u; \boldsymbol{z})$ is L-Lipschitz in both $u$ and $\boldsymbol{z}$.*

We also make the following assumption on the contexts $\boldsymbol{x}_t$. In particular, we do not require the lower bound assumption on the covariance matrix of $F_{\boldsymbol{x}}$, i.e. $\mathbb{E}_{\boldsymbol{x} \sim F_{\boldsymbol{x}}}[\boldsymbol{x}\boldsymbol{x}^\top] \succeq \lambda\boldsymbol{I}$ for some constant $\lambda > 0$, which is common in the literature of online learning with contexts (Ding et al., 2024; Fan et al., 2024; Badanidiyuru et al., 2023). This would require the contexts to be distributed "uniformly" over all directions and can be restrictive.

Additionally, we save the common but strong assumption $f_{\boldsymbol{z}}(\boldsymbol{z}_t) \geq c > 0$ for constant $c$ in the non-parametric literature. We bypass both of them by showing convergence only along the directions of the *realized* contexts and features. More details are deferred until Lemma 3.3.

**Assumption 2** (Stochastic contexts). *The contexts $\boldsymbol{x}_t \in \mathcal{X}$ are generated i.i.d. from an underlying distribution with density $f_{\boldsymbol{x}}$. For simplicity, we assume $\mathcal{X}$ lies in the unit ball, i.e. $\|\boldsymbol{x}_t\|_2 \leq 1$.*

**Assumption 3** (Stochastic features). *The features $\boldsymbol{z}_t \in \mathcal{Z}$ are i.i.d. with density $f_{\boldsymbol{z}}$, which is $L_{\boldsymbol{z}}$-Lipschitz and upper bounded by $\overline{f}_{\boldsymbol{z}}$. Also, $\|\boldsymbol{z}_t\|_2 \leq 1$ lies in the unit ball.*

Under the above model, the expected loss at period $t$ admits the following form.

$$\ell_t(c) = h\int_0^c Q(y - f_*(\boldsymbol{x}_t); \boldsymbol{z}_t)\mathrm{d}y + b\int_c^M 1 - Q(y - f_*(\boldsymbol{x}_t); \boldsymbol{z}_t)\mathrm{d}y. \tag{1}$$

To formalize the learning objective, we compete against the optimal time-varying oracle that has full knowledge of $f_*$, $Q$, and $\{\boldsymbol{z}_t\}_t$. The *dynamic* regret is defined as

$$\mathsf{Reg}(\pi) := \mathbb{E}\left[\sum_{t=1}^T \ell_t(c_t) - \ell_t(c_t^*)\right],$$

---

[1]Note that it is equivalent to assume a Lipschitz demand CDF and to assume a Lipschitz noise CDF.

where $c_t$ is the inventory decision made by the DM's policy $\pi$, and $c_t^* := \arg\min_{c \in [0,M]} \ell_t(c)$ is the optimal decision.

# 3 An Algorithm with Matching Upper Bound

In this section, we introduce and analyze our algorithm for contextual inventory control. Our algorithm builds on two components: (1) an oracle that generates an estimated demand mean conditioned on the context $\boldsymbol{x}_t$ based on historical observations, and (2) a CDF estimator that runs Kernel regression on the noise realizations, but with both *measurement* and *observation* errors. The measurement errors arise from the fact that we do not directly observe the target quantity—the noise $\epsilon_t$—and can only approximate it with the estimated demand mean. The observation errors refer to the deviation of the DM's features $\{\widehat{\boldsymbol{z}}_t\}_t$ from the actual $\{\boldsymbol{z}_t\}_t$, which are taken as input of the noise CDF $Q$.

The algorithm is relatively simple to implement once given the aforementioned oracle. Behind its nice and simple form, however, the core challenge lies in deriving theoretical guarantees under the presence of such measurement errors and the absence of strong assumptions commonly seen in the non-parametric regression literature, e.g. (Fan et al., 2024, Assumption 4.2) that requires a constant lower bound of the feature density $f_{\boldsymbol{z}}$.

## 3.1 Demand Mean Estimation Oracle

For notational convenience, we formulate the estimation oracle for the demand mean $\mathbb{E}_t[D_t]$ as follows. This estimation oracle serves as a black-box in Algorithm 2. In the remaining section, we give two examples and show how to derive the corresponding oracle.

**Assumption 4** (Mean Estimation Oracle). *At each time $t$, the oracle takes in historical observations $\{(\boldsymbol{x}_\tau, D_\tau)\}_{\tau < t}$ and context $\boldsymbol{x}_t$ and outputs an estimated mean $\widehat{D}_t$ such that: with probability at least $1 - T^{-2}$,*

$$|\widehat{D}_t - \mathbb{E}_t[D_t]| \leq \xi_t \text{ for some } \xi_t \geq 0.$$

*Remark* 1. Before proceeding, we remark that our algorithm requires no knowledge of the error bound $\xi_t$. It is only used in a "plug-in" manner in the regret analysis.

### 3.1.1 Linear Demand

First, consider the linear demand model $D_t = \boldsymbol{\theta}_*^\top \boldsymbol{x}_t + \epsilon_t$ with $\|\boldsymbol{\theta}_*\|_2 \leq 1$. Albeit appearing simple, linear models have drawn wide attention and efforts in online learning literature, such as contextual bandits (Abbasi-Yadkori et al., 2011; Chu et al., 2011), ridge bandits (Rajaraman et al., 2024), and online advertising (Badanidiyuru et al., 2023; Wen et al., 2025). The recent work (Ding et al., 2024) in inventory control also focused on this linear demand model.

An oracle is given by a Ridge regression on the data $\{(\boldsymbol{x}_\tau, D_\tau)\}_{\tau < t}$. The closed-form solution is

$$\widehat{\boldsymbol{\theta}}_t = \boldsymbol{A}_t^{-1} \boldsymbol{b}_t \tag{2}$$

where $\boldsymbol{A}_t = \boldsymbol{I} + \sum_{\tau < t} \boldsymbol{x}_\tau \boldsymbol{x}_\tau^\top \in \mathbb{R}^{d \times d}$ and $\boldsymbol{b}_t = \sum_{\tau < t} D_\tau \boldsymbol{x}_\tau \in \mathbb{R}^d$. Thanks to existing results in linear contextual bandits, the estimation error conditioned on the new context $\boldsymbol{x}_t$ is bounded as follows:

**Lemma 3.1.** *Let $\beta = \sqrt{\log(2T)} + 1$. With probability at least $1 - T^{-2}$, it holds that*

$$|\widehat{\boldsymbol{\theta}}_t^\top \boldsymbol{x}_t - \boldsymbol{\theta}_*^\top \boldsymbol{x}_t| \leq \beta \|\boldsymbol{x}_t\|_{\boldsymbol{A}_t^{-1}}.$$

*Moreover, for any $t' \in [T]$, it holds that $\sum_{t=1}^{t'} \|\boldsymbol{x}_t\|_{\boldsymbol{A}_t^{-1}}^2 \leq 2d \log(t')$.*

The error bound $\beta \|\boldsymbol{x}_t\|_{\boldsymbol{A}_t^{-1}}$ at time $t$ depends on how well $\widehat{\boldsymbol{\theta}}_t$ learns along the direction of $\boldsymbol{x}_t$, captured by the interaction between $\boldsymbol{x}_t$ and $\boldsymbol{A}_t$. While it is not monotone, fortunately, the sum of these errors can be bounded by an elliptical potential lemma; see e.g. Abbasi-Yadkori et al. (2011); Chu et al. (2011). We give more details in Appendix B for completeness.

---

**Algorithm 1:** Training Routine for Neural Network

**1 Input:** Epochs $J$, data $\{(\boldsymbol{x}_\tau, D_\tau)\}_{\tau < t}$, regularization $\lambda > 0$, step size $\eta > 0$.

**2 Initialize:** Set initial parameter $\widehat{\boldsymbol{\theta}}_t^{(0)}$ according to Appendix C.

**3** Define loss $\mathcal{L}(\boldsymbol{\theta}) \coloneqq \frac{1}{2} \sum_{\tau < t} (D_\tau - f(\boldsymbol{x}_\tau; \boldsymbol{\theta}))^2 + \lambda \frac{w}{2} \|\boldsymbol{\theta}\|_2$.

**4 for** $j = 1, 2, \ldots, J$ **do**

**5** $\quad$ Update $\widehat{\boldsymbol{\theta}}_t^{(j)} \leftarrow \widehat{\boldsymbol{\theta}}_t^{(j-1)} - \eta \nabla \mathcal{L}(\widehat{\boldsymbol{\theta}}_t^{(j-1)})$.

**6** Return $\widehat{\boldsymbol{\theta}}_t \leftarrow \widehat{\boldsymbol{\theta}}_t^{(J)}$.

---

### 3.1.2 NONLINEAR DEMAND VIA NEURAL NETWORKS

Now we address a more general demand $D_t = f_*(\boldsymbol{x}_t) + \epsilon_t$ where $f_* : \mathbb{R}^d \to \mathbb{R}$ is an arbitrary nonlinear function. To the best of our knowledge, such a general nonlinear formulation has not been explored in the online inventory literature. This section proposes to learn $f_*$ through an over-parametrized neural network and takes inspiration from `NeuralUCB` (Zhou et al., 2020) to develop an error bound. Specifically, a network with width $w > 0$ and depth $K > 0$ is defined as[2]

$$f(\boldsymbol{x}; \boldsymbol{\theta}) = \sqrt{w} \boldsymbol{W}_K \sigma(\boldsymbol{W}_{K-1} \sigma(\cdots \sigma(\boldsymbol{W}_1 \boldsymbol{x}) \cdots)) \tag{3}$$

where $\sigma$ denotes the element-wise ReLU function, weight matrices $\boldsymbol{W}_1 \in \mathbb{R}^{w \times d}$, $\boldsymbol{W}_K \in \mathbb{R}^{1 \times w}$, and $\boldsymbol{W}_k \in \mathbb{R}^{w \times w}$ for $k = 2, \ldots, K-1$.

Due to space limit, we defer the details of the neural analysis to Appendix C. At a high level, the following high-probability error bound relies on the idea of Neural Tangent Kernel (NTK) that when $w$ is sufficiently large, i.e. when the network is over-parametrized, it becomes approximately linear in the parameter space (Jacot et al., 2018). While it remains a central challenge to derive analogous results for smaller networks in current deep learning theory, we will empirically validate in Section 5 that small networks are typically sufficient for capturing demand mean and yielding a vanishing regret in practice.

**Lemma 3.2** (Informal). *Let $\widehat{\boldsymbol{\theta}}_t$ be trained using Algorithm 1 at time $t$. When we use sufficiently large epochs $J = \widetilde{\Omega}(TK/\lambda)$, width $w = \widetilde{\Omega}(poly(T, K, \lambda^{-1}))$, and small step size $\eta = \widetilde{O}((wTK + w\lambda)^{-1})$, with probability at least $1 - T^{-2}$, we have*

$$|f(\boldsymbol{x}_t; \widehat{\boldsymbol{\theta}}_t) - f_*(\boldsymbol{x}_t)| \le \xi_t$$

*for some $\xi_t \ge 0$ that satisfies: For any $t' \in [T]$, it holds that $\sum_{t=1}^{t'} \xi_t^2 = \widetilde{O}(\widetilde{d} \log(t'))$ where the factor $\widetilde{d}$ depends on a function norm of the ground truth $f_*$ and the effective dimension of the NTK matrix of $f(\cdot; \boldsymbol{\theta})$; see Appendix C.3 for details.*

### 3.2 ALGORITHM

Before going into the details of the more involved non-parametric estimation, we present our algorithm to give the readers an idea of how the mean estimation oracles are used and how their error complicates the non-parametric regression. At each time $t$, Algorithm 2 computes an estimated demand mean using a given oracle that satisfies Assumption 4 and an estimated noise CDF $\widetilde{Q}_t$ that will be discussed in the next section. Then the algorithm computes a surrogate loss

$$\widehat{\ell}_t(c) = h \int_0^c \widetilde{Q}_t(y - \widehat{D}_t)\mathrm{d}y + b \int_c^M \left[1 - \widetilde{Q}_t(y - \widehat{D}_t)\right] \mathrm{d}y \tag{4}$$

and simply orders the inventory level up to the maximizer with respect to $\widehat{\ell}_t$.

Nonetheless, classical non-parametric regression builds on the *precise* observations of the covariates $\{\boldsymbol{z}_\tau\}_{\tau < t}$ and the variable realizations $\{\epsilon_\tau\}_{\tau < t}$, none of which is available in our case. This introduces the core challenge in the next section.

---

[2]Note that if the input $\boldsymbol{x}$ is concatenated with a constant entry 1, this formulation subsumes the neural networks with biases, so their representation power remains the same when trained near-optimally.

---

**Algorithm 2:** Contextual online inventory control

**1 Input:** Time horizon $T$, choice space $\mathcal{C} = [0, M]$, unit costs $b, h > 0$, Lipschitz constant $L > 0$,
  a mean estimation oracle $\mathcal{O}$.

**2 for** $t = 1, 2, \ldots, T$ **do**

**3**   Observe the context vector $\boldsymbol{x}_t \in \mathbb{R}^d$ and the transformed features $\widehat{\boldsymbol{z}}_t \in \mathbb{R}^p$.

**4**   Generate estimated demand mean $\widehat{D}_t \leftarrow \mathcal{O}(\{(\boldsymbol{x}_\tau, D_\tau)\}_{\tau < t}; \boldsymbol{x}_t)$.

**5**   Estimate $\widetilde{Q}_t$ from $\{(\widehat{\epsilon}_\tau, \widehat{\boldsymbol{z}}_\tau)\}_{\tau < t}$ and $\widehat{\boldsymbol{z}}_t$ via the NW estimator in equation 6.

**6**   **for** *inventory choice $c \in \mathcal{C}$* **do**

**7**     Compute the loss estimator $\widehat{\ell}_t(c)$ as in equation 4.

**8**   Order the inventory quantity $c_t \leftarrow \arg\max_{c \in \mathcal{C}} \widehat{\ell}_t(c)$.

**9**   Observe the realized demand $D_t$.

**10**  Compute the estimated noise term $\widehat{\epsilon}_t \leftarrow D_t - \widehat{D}_t$.

---

### 3.3 NON-PARAMETRIC REGRESSION WITH ERRORS

For the purpose of demonstration, imagine that the DM has access to the precise features $\{\boldsymbol{z}_\tau\}_{\tau \leq t}$ at each time $t$. For each $u \in \mathcal{C}$ and $\boldsymbol{z}_t \in \mathcal{Z}$, consider the Nadaraya-Watson (NW) kernel regression method:

$$\widehat{Q}_t(u) \equiv \widehat{Q}(u; \boldsymbol{z}_t) := \sum_{\tau=1}^{t-1} \frac{K\left(\frac{\boldsymbol{z}_\tau - \boldsymbol{z}_t}{a_t}\right) \mathbb{1}[\widehat{\epsilon}_\tau \leq u]}{\sum_{\tau < t} K\left(\frac{\boldsymbol{z}_\tau - \boldsymbol{z}_t}{a_t}\right)} \tag{5}$$

where $K$ is a smoothing kernel (e.g. Gaussian kernel) and $a_t > 0$ the bandwidth parameter. As previously discussed, we still face the following technical difficulties: (1) only an approximation $\widehat{\epsilon}_\tau$ is available for the target quantity $\epsilon_\tau$, and (2) without imposing a constant lower bound on $f_{\boldsymbol{z}}$, it is impossible to guarantee a uniform convergence for every new $\boldsymbol{z}_t$.

Let $f_{a_t}(\boldsymbol{z}) = \frac{1}{t-1} \sum_{\tau \in [t-1]} K_{a_t}(\boldsymbol{z}_\tau - \boldsymbol{z})$ be the kernel-smoothed estimator for $f_{\boldsymbol{z}}(\boldsymbol{z})$ with the rescaled kernel $K_{a_t}$. The following result provides a point-wise error bound that depends on the performance of the mean estimation oracle, i.e., on its estimation error appearing in $|\widehat{\epsilon}_\tau - \epsilon_\tau|$. The proof is deferred to Appendix D.1.

**Lemma 3.3.** *Suppose Assumptions 1–3 hold, $t > 1$, and $|\widehat{\epsilon}_\tau - \epsilon_\tau| \leq \xi_\tau$ for every $\tau \in [t-1]$. Then with probability at least $1 - T^{-2}$,*

$$\left| \widehat{Q}(u; \boldsymbol{z}) - Q(u; \boldsymbol{z}) \right| \leq \frac{C_0 \sqrt{\log(T)}}{f_{a_t}(\boldsymbol{z})} \left( L \frac{\overline{\xi}_t}{t} + t^{-\frac{1}{p+2}} \right)$$

*for every $u \in \mathcal{C}$ and $\boldsymbol{z} \in \mathcal{Z}$, with the constant $C_0$ depends on $K$ and $\overline{f}_{\boldsymbol{z}}$. Here the bandwidth is set to $a_t = t^{-\frac{1}{p+2}} p^{\frac{2}{p+2}}$ and $\overline{\xi}_t := \sum_{\tau < t} \xi_\tau$.*

To avoid the strong and often unrealistic assumptions on the density $f_{\boldsymbol{z}}$ commonly imposed in the literature, it is essential to develop bounds that adapt to how well the distribution has been learned around $\boldsymbol{z}_t$ at time $t$. This adaptation is reflected in Lemma 3.3, where the term $f_{a_t}(\boldsymbol{z})$ serves to approximate $f_{\boldsymbol{z}}(\boldsymbol{z})$. When the upcoming feature $\boldsymbol{z}_t$ lies in a low-probability region, the approximation $f_{a_t}(\boldsymbol{z}_t)$ may deviate significantly from $f_{\boldsymbol{z}}(\boldsymbol{z}_t)$, potentially yielding loose or even vacuous error bounds. When $f_{\boldsymbol{z}}(\boldsymbol{z}_t)$ is large, the bound remains uniform in $t$. The key insight is that problematic bounds can only occur with small probability at each round, so the cumulative regret is ultimately governed by the learning performance in the high-probability regions.

Now we revisit the setting where the DM only observes approximate features $\{\widehat{\boldsymbol{z}}_\tau\}_{\tau \leq t}$. Denote $\|\widehat{\boldsymbol{z}}_\tau - \boldsymbol{z}_\tau\|_2 \leq \delta_\tau$ for some error bound $\delta_\tau$ that is $\mathcal{F}_\tau$-measurable. Consider the following surrogate of equation 5:

$$\widetilde{Q}_t(u) \equiv \widetilde{Q}(u; \widehat{\boldsymbol{z}}_t) := \sum_{\tau=1}^{t-1} \frac{K\left(\frac{\widehat{\boldsymbol{z}}_\tau - \widehat{\boldsymbol{z}}_t}{a_t}\right) \mathbb{1}[\widehat{\epsilon}_\tau \leq u]}{\sum_{\tau < t} K\left(\frac{\widehat{\boldsymbol{z}}_\tau - \widehat{\boldsymbol{z}}_t}{a_t}\right)} \tag{6}$$

To understand the performance of this estimator, we build on Lemma 3.3 and bound the error introduced by the observation errors $\delta_\tau$. The proof can be found in Appendix D.3.

**Lemma 3.4.** *Suppose Assumptions 1–3 hold, $t > 1$, $|\widehat{\epsilon}_\tau - \epsilon_\tau| \leq \xi_\tau$, and $\|\widehat{z}_\tau - z_\tau\|_2 \leq \delta_\tau$ for every $\tau \in [t-1]$. Then with probability at least $1 - 4T^{-2}$,*

$$\left| \widetilde{Q}(u; \widehat{z}) - \widehat{Q}_t(u; z) \right| \leq \frac{C_1 \sqrt{\log(T)}}{f_{a_t}(z)} \left( \frac{1}{t-1} \sum_{\tau < t} (1 + \xi_\tau)(\delta_\tau + \delta_t) + t^{-\frac{1}{p+2}} \right)$$

*for every $u \in \mathcal{C}$, $z \in \mathcal{Z}$, and $\|\widehat{z} - z\|_2 \leq \delta_t$. The constant $C_1 > 0$ depends on the kernel $K$, $L$, $L_z$, and $\overline{f}_z$. Here the bandwidth is set to $a_t = t^{-\frac{1}{p+2}} p^{\frac{2}{p+2}}$.*

### 3.4 REGRET UPPER BOUND ANALYSIS

The following theorem establishes the regret upper bound of Algorithm 2. For the sole purpose of readability, we impose an extra assumption on the convergence rate of the mean oracle and the observation errors $\|\widehat{z}_\tau - z_\tau\|_2$. Note this assumption is satisfied by both examples in Section 3.1: (1) in the linear model, $\alpha = \widetilde{O}(d)$; (2) in the nonlinear model $\alpha = \widetilde{O}(\widetilde{d})$ that depends on the truth $f_*$ and the NTK matrix. Our results can be easily extended to general $\{\xi_t\}$ and $\{\widehat{z}_t\}$.

**Theorem 1.** *Suppose the DM has an oracle satisfying Assumption 4 such that $\sum_{\tau=1}^t \xi_\tau^2 \leq \alpha \log(t)$ for every $t \in [T]$ for some $\alpha = o(t)$. Also suppose $\sum_{\tau=1}^t \|\widehat{z}_\tau - z_\tau\|_2^2 \leq \alpha \log(t)$ for every $t \in [T]$. Under Assumptions 1–3 and setting bandwidth $a_t = t^{-\frac{1}{p+2}} p^{\frac{2}{p+2}}$, we have*

$$\mathsf{Reg}(\mathsf{Alg}\ 2) = O\left( (b+h)M(L+1)\log(T)\left( \sqrt{\alpha T} + T^{\frac{p+1}{p+2}} \right) \right) = \widetilde{O}\left( \sqrt{\alpha T} + T^{\frac{p+1}{p+2}} \right).$$

The rest of this section is devoted to the proof of Theorem 1. Without loss of generality (WLOG), we assume the high-probability events in Assumption 4 (e.g., by Lemma 3.1 or 3.2), in Lemma 3.3, and in Lemma 3.4 hold. By Cauchy-Schwartz inequality, $\sum_{\tau=1}^t \xi_\tau \leq \sqrt{\alpha t \log(t)}$. Then at each time $t$, by Lemma 3.3 and 3.4 and our assumptions, it holds that

$$\left| \widehat{\ell}_t(c) - \ell_t(c) \right| \leq (b+h)M\left( L\xi_t + 2 \max_{u \in \mathcal{C}} \left| \widetilde{Q}_t(u) - Q_t(u) \right| \right)$$

$$= O\left( (b+h)ML\left( \xi_t + \frac{\sqrt{\log(T)}}{f_{a_t}(z_t)}\left( t^{-\frac{1}{p+2}} + \sqrt{\frac{\alpha}{t}} + \|\widehat{z}_t - z_t\|_2 \right) \right) \right) =: \omega_t$$

where we introduce the notation $\omega_t$ to compactly write the error bound for the loss estimator. Consequently, the instantaneous regret of selecting the greedy maximizer at time $t$ is bounded by

$$\max_{c \in \mathcal{C}} \ell_t(c) = \ell_t(c_t^*) \leq \widehat{\ell}_t(c_t^*) + \omega_t \leq \max_{c \in \mathcal{C}} \widehat{\ell}_t(c) + \omega_t \leq \ell_t(c_t) + 2\omega_t.$$

The regret is then bounded as the sum of the instantaneous errors

$$\mathsf{Reg}(\mathsf{Alg}\ 2) = \mathbb{E}\left[ \sum_{t=1}^T \ell_t(c_t) - \ell_t(c_t^*) \right] \leq 2\mathbb{E}\left[ \sum_{t=1}^T \min\{(b+h)M, \omega_t\} \right]. \tag{7}$$

To handle the term $f_{a_t}(z_t)^{-1}$, by Lemma D.2, Lemma D.3, and bandwidth choice $a_t = t^{-\frac{1}{p+2}} p^{\frac{2}{p+2}}$ in as in Lemma 3.4, we have $f_{a_t}(z_t) \geq \frac{1}{2} f_z(z_t)$ when $f_z(z_t) \geq 2C_0 \log(T) t^{-\frac{1}{p+2}}$, where $C_0$ is the constant in the lemmas that depend on the kernel $K$. By Assumption 3, $z_t$ are i.i.d.. Define an event indicator $I_t = \mathbb{1}[f_z(z_t) \geq 2C_0 \log(T) t^{-\frac{1}{p+2}}]$. Then for any value $v \in [0, M]$, the following decomposition holds:

$$\mathbb{E}_t\left[ \min\left\{ (b+h)M, \frac{v}{f_{a_t}(z_t)} \right\} \right] \leq \mathbb{E}_t\left[ I_t \frac{v}{f_{a_t}(z_t)} \right] + (b+h)M\mathbb{P}(I_t = 0)$$

$$\leq \mathbb{E}_t\left[ \frac{2v}{f_z(z_t)} \right] + 2(b+h)MC_0 \log(T) t^{-\frac{1}{p+2}} |\mathcal{Z}|$$

$$= O\left( v + (b+h)M \log(T) t^{-\frac{1}{p+2}} \right)$$

where the last line uses $\mathbb{E}_t[1/f_{\boldsymbol{z}}(\boldsymbol{z}_t)] = \int_{\mathcal{Z}} \mathrm{d}(\boldsymbol{z}_t) = |\mathcal{Z}| = O(1)$. Combining this with equation 7 leads to the desired bound with some constant $C' > 0$:

$$\text{Reg(Alg 2)} \leq C'(b+h)M(L+1)\sum_{t=1}^{T}\left(\xi_t + \sqrt{\log(T)}\left(t^{-\frac{1}{p+2}} + \sqrt{\frac{\alpha}{t}} + \|\widehat{\boldsymbol{z}}_t - \boldsymbol{z}_t\|_2\right)\right)$$

$$= O\left((b+h)M(L+1)\log(T)\left(\sqrt{\alpha T} + T^{\frac{p+1}{p+2}}\right)\right)$$

where we apply the elementary inequalities $\sum_{t=1}^{T}\frac{1}{\sqrt{t}} = O(\sqrt{T})$ and $\sum_{t=1}^{T}t^{-\frac{1}{p+2}} = O(T^{\frac{p+1}{p+2}})$.

### 3.5 Breaking the Curse of Dimensionality under Benign Distributions

The term $T^{\frac{p+1}{p+2}}$ showcases the inherent difficulty in non-parametric regression and is generally not improvable without additional regularity conditions. In this section, we provide an arguably mild regularity condition on $f_{\boldsymbol{z}}$, under which we achieve $\widetilde{O}(\sqrt{\alpha T} + p\sqrt{T})$ regret.

**Assumption 5.** *There exist constants $c_{FT}, C_{FT}, \omega > 0$ such that for every $\boldsymbol{v} \in \mathbb{R}^p$ and $u \in \mathcal{C}$,*

$$\max\{|\mathcal{T}[Q(u;\cdot)f_{\boldsymbol{z}}(\cdot)](\boldsymbol{v})|, |\mathcal{T}[f_{\boldsymbol{z}}](\boldsymbol{v})|\} \leq C_{FT}\exp(-c_{FT}\|\boldsymbol{v}\|_2^{\omega})$$

*where $\mathcal{T}[f](\boldsymbol{v}) = \int_{\mathbb{R}^p} f(\boldsymbol{z})e^{-i\boldsymbol{v}^{\top}\boldsymbol{z}}\mathrm{d}\boldsymbol{z}$ denotes the Fourier Transform of $f$.*

Assumption 5 asks the Fourier coefficients of the feature density $f_{\boldsymbol{z}}(\boldsymbol{z})$ and the unconditional probability (for fixed $u \in [0, M]$) $Q(u;\boldsymbol{z})f_{\boldsymbol{z}}(\boldsymbol{z})$ to decay at a fast rate, which is typically stronger than infinite smoothness. Then with an infinitely smooth kernel (that does not depend on the constants in Assumption 5; see equation 19), we can guarantee an improved bound for the NW estimator in equation 6:

**Lemma 3.5.** *Under same assumptions as Lemma 3.4 and Assumption 5, there exists a kernel $K$ such that: with probability at least $1 - 4T^{-2}$,*

$$\left|\widetilde{Q}(u;\boldsymbol{z}) - Q(u;\boldsymbol{z})\right| \leq \frac{\gamma'}{f_{a_t}(\boldsymbol{z})}\left(L\frac{\sum_{\tau < t}\xi_\tau}{t} + \frac{1}{t-1}\sum_{\tau < t}(1+\xi_\tau)(\delta_\tau + \delta_t) + \frac{p}{\sqrt{t}}\right)$$

*for every $u \in \mathcal{C}$ and $\boldsymbol{z} \in \mathcal{Z}$, with $\gamma' = O\left(\log(T)^{\frac{p}{\omega}}\log\log(T)^{\frac{1}{2}}\right)$.*

The proof is deferred to Appendix D.2. Consequently, we arrive at the following improved regret guarantee for Algorithm 2. The proof is the same as Section 3.4, except for replacing Lemma 3.3 and 3.4 by Lemma 3.5, and hence is omitted.

**Theorem 2.** *Under same assumptions as Theorem 1 and additionally Assumption 5, it holds that* $\text{Reg(Alg 2)} = \widetilde{O}\left(\sqrt{\alpha T} + p\sqrt{T}\right)$.

## 4 Minimax Regret Lower Bound

To complement our upper bound result in Theorem 1, we provide a minimax lower bound when the underlying environment admits the *linear* model in Section 3.1.1. Note the lower bound in Theorem 3 holds for *any* policy $\pi$, regardless of whether the DM uses a linear oracle or a (over-complicated) nonlinear one in the learning algorithm. The proof is left to Appendix A.

**Theorem 3** (Lower bound). *Suppose the DM perfectly observes $\boldsymbol{z}_t$ at time $t$. When $T \geq d^2$, we have*

$$\inf_{\pi}\sup_{\boldsymbol{\theta}_*, Q, f_{\boldsymbol{z}}, f_{\boldsymbol{x}}} \text{Reg}(\pi) = \Omega\left((b+h)M\left(\sqrt{dT} + T^{\frac{p+1}{p+2}}\right)\right),$$

*where inf is taken over all possible policies, and sup is taken over the problem parameters that satisfy $\|\boldsymbol{\theta}_*\|_2 \leq 1$ and Assumptions 1–3.*

Recall that when the noise CDF $Q_t \equiv Q$ is time-invariant, the regret is $\widetilde{O}(\sqrt{T})$, given by the OSGD algorithm (Hazan et al., 2016; Ding et al., 2024). Thus, as soon as the shape of the demand distribution is not time-invariant, the online inventory control problem becomes fundamentally harder.[3]

---

[3]When $Q_t$ is invariant, the optimal solution $c_t^*$ is linear in $\boldsymbol{x}_t$, so the DM can compute its gradient and learn simultaneously in $d$ directions. In the more general case, however, this convenient structure breaks down.

## 5 EXPERIMENTS

We conclude the paper with a series of numerical experiments, where we test our algorithm on both synthetic datasets and real-world datasets against the benchmark OSGD in online inventory control.

**Synthetic Datasets** We first compare the Algorithm 2 to that of the OSGD. Due to space limits, we leave all the setup details and results to Appendix E.1, which highlights the superior performance of our algorithm compared to OSGD under various synthetic settings, including heteroskedastic noise and nonlinear expected demand.

**Real-world Dataset** We evaluate our algorithm on real-world datasets from the M5 Forecasting-Accuracy dataset (Howard et al., 2020), which includes the sales and feature data of more than 30,000 items during 1,941 days. We compare the OSGD to our Algorithm 2 with the ridge regression (assuming linear demand) and neural network estimators (assuming nonlinear demand). Details about data description and algorithm parameters are left to Appendix E.2. In Figure (a) below, we consider 40 items with the most nonzero selling periods (all of which happen to be food) and present the statistics of the cumulative loss of the three algorithms. To gain more insights, Figure (b) and (c) plot the growth of the cumulative regret for each of the three algorithms for two distinct items.

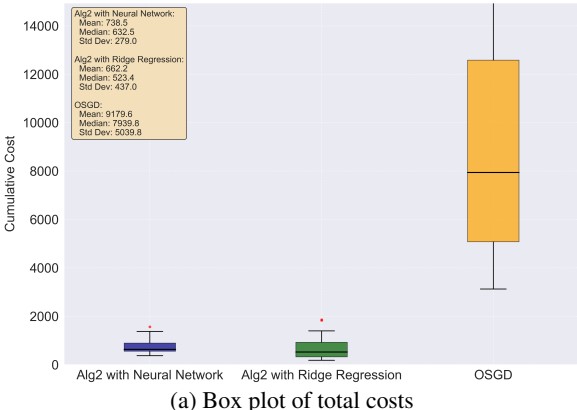

(a) Box plot of total costs

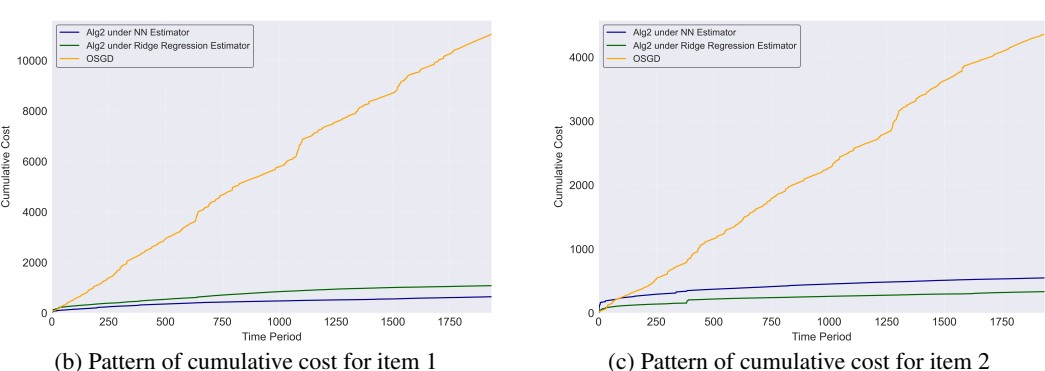

(b) Pattern of cumulative cost for item 1      (c) Pattern of cumulative cost for item 2

Our Algorithm consistently outperforms OSGD on the tested items. Due to the complex heteroskedasticity in real data, OSGD does not converge within the given time horizon in most cases. In terms of the two demand mean estimators we used in Algorithm 2, the linear model using ridge regression achieves a relatively lower average cumulative cost over the 40 items of interest, yet exhibits a larger variance. This indicates that the complexity of real-world sales is beyond the linear model, even after heteroskedasticity is addressed. In turn, it highlights the need for nonlinear treatment in online contextual inventory control as is done in our work, which has been largely overlooked in the existing (in particular, theory-focused) literature. In contrast, despite the simple two-layer architecture, neural networks have demonstrated robust and consistent performance.

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

# A    PROOFS FOR LOWER BOUND IN SECTION 4

To clarify the inherent statistical complexity of the problem, specifically, where do $\Omega(\sqrt{dT})$ and $\Omega(T^{\frac{p+1}{p+2}})$ come from, we introduce the following two lemmas.

**Lemma A.1.** *Consider $p = 1$. Let $z_t = \beta^\top x_t$ with a simple $\beta = [\frac{1}{d}, \frac{2}{d}, \ldots, 1]^\top$ for every time t, and let $f_x$ be uniform over the canonical basis $\{e_1, \ldots, e_d\}$. When $T \geq d^2$:*

$$\inf_\pi \sup_{\theta_*, Q} \mathsf{Reg}(\pi) = \Omega((b + h)M\sqrt{dT}).$$

**Lemma A.2.** *WLOG, consider $d = p$ and $z_t = x_t$. Assume that $\theta_*$ is perfectly known. The following lower bound holds:*

$$\inf_\pi \sup_{\theta_*, Q} \mathsf{Reg}(\pi) = \Omega\left((b + h)MT^{\frac{p+1}{p+2}}\right).$$

Note for $p < d$, it is straightforward to apply the analysis of Lemma A.2 by having $d - p$ entries of the context $x$ to be zero.

## A.1    PROOF OF $\Omega(\sqrt{dT})$

The lower bound of regret $\Omega(\sqrt{dT})$ comes from jointly estimating $\theta_*$ and $Q_t$ when the true noise distribution $Q_t$ is also allowed to depend on $x_t$ (even in a quite simple way and $p = 1$). This dependence structure leads to the extra complexity of $\sqrt{d}$ term.

*Proof of Lemma A.1.* We construct the space of linear coefficient vectors $\theta_* \in \Theta$, with $\Theta = \mathsf{Unif}\left(\{-\Delta, \Delta\}^d\right)$ for some small $\Delta > 0$ that will be determined later. Let $x^j := (0, \cdots, 0, 1, 0, \cdots, 0) = e_j$, where all except the $j$-th element are 0, and $x_t$ is uniformly sampled from $\{x^1, x^2, \cdots, x^d\}$. Write $\theta_*^\top x_t =: \mu_t \in \{\pm\Delta\}$ for simplicity. The noise distribution $Q_t$ is constructed as follows:

$$Q_t(y) = \begin{cases} \left(\dfrac{4\rho}{M} + \dfrac{4\mu_t}{M}\right) \cdot (y + \mu_t), & \text{for } -\mu_t \leq y < -\mu_t + \dfrac{M}{4}, \\[3mm] \rho + \mu_t, & \text{for } -\mu_t + \dfrac{M}{4} \leq y < -\mu_t + \dfrac{3M}{4}, \\[3mm] \left(\dfrac{4(1-\beta)}{M} - \dfrac{4\mu_t}{M}\right) \cdot \left(y + \mu_t - \dfrac{3M}{4}\right) + \rho + \mu_t, & \text{for } -\mu_t + \dfrac{3M}{4} \leq y < -\mu_t + M, \\[3mm] 1, & \text{for } y = -\mu_t + M. \end{cases}$$

Note given the structure of $z_t = \beta^\top x_t \in \{\frac{1}{d}, \ldots, 1\}$, $Q_t$ is effectively observing $x_t$, and hence is allowed to depend on $\mu_t$ as above. It can be easily verified that such a construction satisfies our Assumptions 1-3, except for the zero-mean property of $\epsilon_t$ which can be achieved through scaling (e.g., through tuning the first component of $\theta_*$ and $x_t$). We do not present it and still use $Q_t$ for simplicity. Now the conditional demand CDF (where our observations are sampled from, and we slightly abuse the notation $Q$ also to denote the demand distribution as it is only a shift of the noise distribution) can be written as:

$$G_t(y) = \begin{cases} \left(\dfrac{4\rho}{M} + \dfrac{4\theta_*^\top x_t}{M}\right) \cdot y, & \text{for } 0 \leq y < \dfrac{M}{4}, \\[3mm] \rho + \theta_*^\top x_t, & \text{for } \dfrac{M}{4} \leq y < \dfrac{3M}{4}, \\[3mm] \left(\dfrac{4(1-\beta)}{M} - \dfrac{4\theta_*^\top x_t}{M}\right) \cdot \left(y - \dfrac{3M}{4}\right) + \rho + \theta_*^\top x_t, & \text{for } \dfrac{3M}{4} \leq y < M, \\[3mm] 1, & \text{for } y = M. \end{cases}$$

Conditioned on any $x_t$, there are two possible demand distributions in total: when $\theta_*^\top x_t = \Delta$, we denote the corresponding noise CDF as $Q_\Delta(y)$, and we have the optimal solution $c^*_{Q_\Delta} \in (0, M/4)$;

when $\boldsymbol{\theta}_*^\top \boldsymbol{x}_t = -\Delta$, we denote the corresponding CDF as $Q_{-\Delta}(y)$, and we have the optimal solution $c_{Q_{-\Delta}}^* \in (3M/4, M)$. Now we can lower bound the regret as follows:

$$\sup_{\boldsymbol{\theta}_*, Q} \mathsf{Reg}(\pi) \geq \frac{1}{2} \left\{ \sum_{t=1}^T \mathbb{E}_{\pi, Q_\Delta}[\ell_t(c_t) - \ell_t(c_{Q_\Delta}^*)] + \sum_{t=1}^T \mathbb{E}_{\pi, Q_{-\Delta}}[\ell_t(c_t) - \ell_t(c_{Q_{-\Delta}}^*)] \right\}$$

$$\overset{(i)}{=} \frac{1}{2d} \left\{ \sum_{j=1}^d \left[ \sum_{t=1}^T \mathbb{E}_{\pi, Q_\Delta}[\ell_t(c_t) - \ell_t(c_{Q_\Delta}^*)|\boldsymbol{x}_t = \boldsymbol{x}^j] + \sum_{t=1}^T \mathbb{E}_{\pi, Q_{-\Delta}}[\ell_t(c_t) - \ell_t(c_{Q_{-\Delta}}^*)|\boldsymbol{x}_t = \boldsymbol{x}^j] \right] \right\}$$

$$\overset{(ii)}{=} \frac{1}{2d} \left\{ \sum_{j=1}^d \left[ \sum_{t=1}^T \mathbb{E}_{\pi, Q_{\{\theta_j=\Delta\}}}[\ell_t(c_t) - \ell_t(c_{Q_\Delta}^*)|\boldsymbol{x}_t = \boldsymbol{x}^j] + \sum_{t=1}^T \mathbb{E}_{\pi, Q_{\{\theta_j=-\Delta\}}}[\ell_t(c_t) - \ell_t(c_{Q_{-\Delta}}^*)|\boldsymbol{x}_t = \boldsymbol{x}^j] \right] \right\}$$

$$\overset{(iii)}{\geq} \frac{\Delta M(h+b)}{8d} \cdot \sum_{j=1}^d \sum_{t=1}^T \mathbb{E}_\pi[Q_{\{\theta_j=\Delta\}}(c_t > M/2) + Q_{\{\theta_j=-\Delta\}}(c_t \leq M/2)|\boldsymbol{x}_t = \boldsymbol{x}^j]$$

$$\overset{(iv)}{\geq} \frac{\Delta M(h+b)}{8d} \cdot \sum_{j=1}^d \sum_{t=1}^T \mathbb{E}_\pi[\exp(-t \cdot \mathrm{D}_{\mathrm{KL}}(Q_{\{\theta_j=\Delta\}} \| Q_{\{\theta_j=-\Delta\}}))|\boldsymbol{x}_t = \boldsymbol{x}^j]$$

$$\overset{(v)}{=} \frac{\Delta M(h+b)}{8} \cdot \sum_{j=1}^d \mathbb{E}_{\pi, \boldsymbol{x}} \sum_{t=1}^{T_j} \exp(-t \cdot \mathrm{D}_{\mathrm{KL}}(Q_{\{\theta_j=\Delta\}} \| Q_{\{\theta_j=-\Delta\}})),$$

where (i) comes from the tower property of conditional expectation; (ii) is because conditional on $\boldsymbol{x}_t = \boldsymbol{x}^j$, we must have $\theta_j = \Delta$ and $\theta_j = -\Delta$ for $Q_\Delta$ and $Q_{-\Delta}$, respectively, where we use $\theta_j$ to denote the $j$-th element of $\boldsymbol{\theta}_*$, and we use $Q_{\{\theta_j=\Delta\}}$ and $Q_{\{\theta_j=-\Delta\}}$ to represent the corresponding demand distribution; (iii) directly follows from the analysis in (Zhang et al., 2020, Proposition 1); (iv) comes from the Bretagnolle-Huber Inequality; (v) uses the fact that $T_j := \sum_{t=1}^T \mathbf{1}(\boldsymbol{x}_t = \boldsymbol{x}^j)$ and the sub-problems of learning $\theta_j$ with observations $\boldsymbol{x}_t = \boldsymbol{x}^j$ are *independent*, and that only the $T_j$ observations will contribute to the total KL divergence according to our construction on $\boldsymbol{\theta}_*$ and $\boldsymbol{x}_t$.

For the KL divergence of $\mathrm{D}_{\mathrm{KL}}(Q_{\{\theta_j=\Delta\}} \| Q_{\{\theta_j=-\Delta\}})$, we have:

$$\mathrm{D}_{\mathrm{KL}}(Q_{\{\theta_j=\Delta\}} \| Q_{\{\theta_j=-\Delta\}}) = (\rho + \Delta) \cdot \log \frac{\rho + \Delta}{\rho - \Delta} + (1 - \rho - \Delta) \cdot \log \frac{1 - \rho - \Delta}{1 - \rho + \Delta}$$

$$\leq (2\rho^{-1} + (1-\rho)^{-1} + \rho^{-2}) \cdot \Delta^2,$$

see e.g. analysis of (Besbes & Muharremoglu, 2013, Lemma 4). Let $c_0 := 2\rho^{-1} + (1-\rho)^{-1} + \rho^{-2}$ and taking the result back to the previous inequalities, we have:

$$\sup_{\boldsymbol{\theta}_*, Q, F_{\boldsymbol{x}}} \mathsf{Reg}(\pi) \geq \frac{\Delta M(h+b)}{8} \cdot \sum_{j=1}^d \mathbb{E}_{\pi, \boldsymbol{x}} \sum_{t=1}^{T_j} \exp(-t \cdot \mathrm{D}_{\mathrm{KL}}(Q_{\{\theta_j=\Delta\}} \| Q_{\{\theta_j=-\Delta\}}))$$

$$\geq \frac{\Delta M(h+b)}{8} \cdot \sum_{j=1}^d \mathbb{E}_{\pi, \boldsymbol{x}} \sum_{t=1}^{T_j} \exp(-c_0 t \cdot \Delta^2))$$

$$\geq \frac{\Delta M(h+b)}{8} \cdot \sum_{j=1}^d \mathbb{E}_{\pi, \boldsymbol{x}} \sum_{t=1}^{\min\{T_j, \Delta^{-2}\}} \exp(-c_0)$$

$$\overset{(i)}{\geq} \frac{\Delta M(h+b) \exp(-c_0)}{8} \cdot \sum_{j=1}^d \min\{\mathbb{E}[T_j], \Delta^{-2}\},$$

where (i) comes from $\mathbb{E} \min\{T_j, \Delta^{-2}\} \geq \min\{\mathbb{E}[T_j], \Delta^{-2}\}$. Since $\mathbb{E}[T_j] = \frac{T}{d}$, by setting $\Delta := \sqrt{\frac{d}{T}}$ and $T \geq d^2$ (so that $\|\boldsymbol{\theta}_*\|_2 \leq 1$), we conclude that

$$\sup_{\boldsymbol{\theta}_*, Q, F_{\boldsymbol{x}}} \mathsf{Reg}(\pi) \geq \frac{\exp(-c_0)}{8} \cdot M(b+h)\sqrt{dT} = \Omega((b+h)M\sqrt{dT}).$$

$\square$

## A.2   Proof of $\Omega(T^{\frac{p+1}{p+2}})$

The lower bound of regret $\Omega(T^{\frac{p+1}{p+2}})$ comes from the difficulty in learning a $p$-dimensional non-parametric estimation problem.

*Proof of Lemma A.2.* Assume that the parameter $\boldsymbol{\theta}_*$ is known to the learner and will be specified later. As $z_t$ lies in the $p$-dimensional unit ball, we can construct $\mathcal{G}$ to be any $1/N$-packing of the $p$-dimensional unit ball. By the packing number bound of $\ell_2$ ball, there exists a constant $c_1 > 0$ such that $|\mathcal{G}| = c_1 N^p$. We let $N = \lfloor T^{\frac{1}{p+2}} \rfloor$ and $z_t$ is uniformly sampled from the packing set $\mathcal{G} := \{\boldsymbol{z}^1, \boldsymbol{z}^2, \cdots, \boldsymbol{z}^{c_1 N^p}\}$[4].

Thus, for each vector $\boldsymbol{z}^j$ in $\mathcal{G}$ (which we index by $j = 1, 2, \cdots, c_1 N^p$), consider the following two CDF candidates as similarly defined before in the proof of Lemma A.1:

$$
Q^{(0)}(y) = \begin{cases}
\left( \dfrac{4\rho}{M} + \dfrac{4}{NM} \right) \cdot y, & \text{for } 0 \le y < \dfrac{M}{4}, \\[2ex]
\rho + \dfrac{1}{N}, & \text{for } \dfrac{M}{4} \le y < \dfrac{3M}{4}, \\[2ex]
\left( \dfrac{4(1-\rho)}{M} - \dfrac{4}{NM} \right) \cdot \left( y - \dfrac{3M}{4} \right) + \rho + \dfrac{1}{N}, & \text{for } \dfrac{3M}{4} \le y < M, \\[2ex]
1, & \text{for } y = M.
\end{cases}
$$

and

$$
Q^{(1)}(y) = \begin{cases}
\left( \dfrac{4\rho}{M} - \dfrac{4}{NM} \right) \cdot y, & \text{for } 0 \le y < \dfrac{M}{4}, \\[2ex]
\rho - \dfrac{1}{N}, & \text{for } \dfrac{M}{4} \le y < \dfrac{3M}{4}, \\[2ex]
\left( \dfrac{4(1-\rho)}{M} + \dfrac{4}{NM} \right) \cdot \left( y - \dfrac{3M}{4} \right) + \rho - \dfrac{1}{N}, & \text{for } \dfrac{3M}{4} \le y < M, \\[2ex]
1, & \text{for } y = M,
\end{cases}
$$

Let the underlying noise CDF be $Q(\cdot; z_t = \boldsymbol{z}^j) = Q^{(u_i)}(\cdot)$ for i.i.d. $u_i \sim \text{Bern}(\frac{1}{2})$. It is easy to verify that the construction satisfies our model assumptions, except the zero-mean. Now we can guarantee zero-mean through choosing $\boldsymbol{\theta}_*$, which is known to the DM. Note that the noise distributions are independent for different $\boldsymbol{z}^j \in \mathcal{G}$.

---

[4]We let $c_1 N^p$ to be an integer for notation simplicity, otherwise we can take $\lfloor c_1 N^p \rfloor$ instead.

Similar to the proof of Lemma A.1, we have:

$$\sup_{\boldsymbol{\theta}_*,Q} \mathsf{Reg}(\pi) \geq \frac{1}{2} \left\{ \sum_{t=1}^{T} \mathbb{E}_{\pi,Q^{(0)}}[\ell_t(c_t) - \ell_t(c_{Q^{(0)}}^*)] + \sum_{t=1}^{T} \mathbb{E}_{\pi,Q^{(1)}}[\ell_t(c_t) - \ell_t(c_{Q^{(1)}}^*)] \right\}$$

$$= \frac{1}{2c_1 N^p} \left\{ \sum_{j=1}^{c_1 N^p} \left[ \sum_{t=1}^{T} \mathbb{E}_{\pi,Q^{(0)}}[\ell_t(c_t) - \ell_t(c_{Q^{(0)}}^*)|\boldsymbol{z}_t = \boldsymbol{z}^j] + \sum_{t=1}^{T} \mathbb{E}_{\pi,Q^{(1)}}[\ell_t(c_t) - \ell_t(c_{Q^{(1)}}^*)|\boldsymbol{z}_t = \boldsymbol{z}^j] \right] \right\}$$

$$= \frac{1}{2c_1 N^p} \left\{ \sum_{j=1}^{c_1 N^p} \left[ \sum_{t=1}^{T} \mathbb{E}_{\pi,Q^{(0)}}[\ell_t(c_t) - \ell_t(c_{Q^{(0)}}^*)|\boldsymbol{z}_t = \boldsymbol{z}^j] + \sum_{t=1}^{T} \mathbb{E}_{\pi,Q^{(1)}}[\ell_t(c_t) - \ell_t(c_{Q^{(1)}}^*)|\boldsymbol{z}_t = \boldsymbol{z}^j] \right] \right\}$$

$$\geq \frac{M(h+b)}{N} \frac{1}{c_1 N^p} \sum_{j=1}^{c_1 N^p} \sum_{t=1}^{T} \mathbb{E}_{\pi}[Q^{(0)}(c_t > M/2) + Q^{(1)}(c_t \leq M/2)|\boldsymbol{z}_t = \boldsymbol{z}^j]$$

$$\geq \frac{M(h+b)}{N} \frac{1}{c_1 N^p} \sum_{j=1}^{c_1 N^p} \sum_{t=1}^{T} \mathbb{E}_{\pi}[\exp(-t \cdot \mathrm{D}_{\mathrm{KL}}(Q^{(0)}\|Q^{(1)}))|\boldsymbol{z}_t = \boldsymbol{z}^j]$$

$$= \frac{M(h+b)}{N} \cdot \sum_{j=1}^{c_1 N^p} \mathbb{E}_{\pi,\boldsymbol{z}} \sum_{t=1}^{T_j} \exp(-t \cdot \mathrm{D}_{\mathrm{KL}}(Q^{(0)}\|Q^{(1)}))$$

$$\geq \frac{M(h+b)\exp(-c_0)}{8N} \cdot \sum_{j=1}^{c_1 N^p} \min\{\mathbb{E}[T_j], N^2\},$$

where $c_0 = 2\rho^{-1} + (1-\rho)^{-1} + \rho^{-2}$, $T_j := \sum_{t=1}^{T} \mathbf{1}(\boldsymbol{z}_t = \boldsymbol{z}^j)$ is defined similarly as in the previous proof, and $\mathbb{E}[T_j] = \frac{T}{c_1 N^p} \geq \frac{1}{c_1} T^{\frac{2}{p+2}}$. We conclude that

$$\sup_{\boldsymbol{\theta}_*,Q,F_{\boldsymbol{x}}} \mathsf{Reg}(\pi) \geq \frac{\exp(-c_0)}{8} \cdot (b+h)MT^{\frac{p+1}{p+2}} = \Omega\left((b+h)MT^{\frac{p+1}{p+2}}\right).$$

$\square$

## B  LINEAR DEMAND ESTIMATION ORACLE

For the sake of completeness, we present the proof for Lemma 3.1 here.

**Lemma B.1** (Restatement of Lemma 3.1). *Follow the notations in Section 3.1.1. With probability at least $1 - T^{-2}$, it holds that*

$$\left|\widehat{\boldsymbol{\theta}}_t^\top \boldsymbol{x}_t - \boldsymbol{\theta}_*^\top \boldsymbol{x}_t\right| \leq (\sqrt{\log(2T)} + 1)\|\boldsymbol{x}_t\|_{\boldsymbol{A}_t^{-1}}.$$

*Proof.* Denote $\mathbf{U}_t = [\boldsymbol{x}_\tau]_{\tau<t} \in \mathbb{R}^{d\times(t-1)}$ and $\mathbf{V}_t = [D_\tau]_{\tau<t} \in \mathbb{R}^{1\times(t-1)}$. Recall the definition

$$\boldsymbol{A}_t = \boldsymbol{I} + \mathbf{U}_t \mathbf{U}_t^\top, \qquad \boldsymbol{b}_t = \mathbf{U}_t \mathbf{V}_t^\top.$$

Then we have

$$\widehat{\boldsymbol{\theta}}_t^\top \boldsymbol{x}_t - \boldsymbol{\theta}_*^\top \boldsymbol{x}_t = \boldsymbol{x}_t^\top \boldsymbol{A}_t^{-1} \boldsymbol{b}_t - \boldsymbol{x}_t^\top \boldsymbol{A}_t^{-1}(\boldsymbol{I} + \mathbf{U}_t \mathbf{U}_t^\top)\boldsymbol{\theta}_*$$

$$= \boldsymbol{x}_t^\top \boldsymbol{A}_t^{-1} \mathbf{U}_t(\mathbf{V}_t^\top - \mathbf{U}_t^\top \boldsymbol{\theta}_*) - \boldsymbol{x}_t^\top \boldsymbol{A}_t^{-1} \boldsymbol{\theta}_*.$$

Given $\|\boldsymbol{\theta}_*\|_2 \leq 1$, it follows that

$$\left|\widehat{\boldsymbol{\theta}}_t^\top \boldsymbol{x}_t - \boldsymbol{\theta}_*^\top \boldsymbol{x}_t\right| \leq \left|\boldsymbol{x}_t^\top \boldsymbol{A}_t^{-1} \mathbf{U}_t(\mathbf{V}_t^\top - \mathbf{U}_t^\top \boldsymbol{\theta}_*)\right| + \|\boldsymbol{x}_t^\top \boldsymbol{A}_t^{-1}\|_2. \tag{8}$$

Since $(D_\tau)_{\tau<t}$ are conditionally independent, we have $\mathbb{E}\big[\mathbf{V}_{t,1}^\top|(\boldsymbol{x}_\tau)_{\tau<t}\big] = \mathbf{U}_{t,1}^\top\boldsymbol{\theta}_*$. Thus by Azuma-Hoeffding's inequality (Azuma, 1967; Alon & Spencer, 2016),

$$P\Big(\big|\boldsymbol{x}_t^\top\boldsymbol{A}_t^{-1}\mathbf{U}_t\big(\mathbf{V}_t - \mathbf{U}_t^\top\boldsymbol{\theta}_*\big)\big| \geq \sqrt{\log(2T)}\|\boldsymbol{x}_t\|_{\boldsymbol{A}_t^{-1}}\Big) \leq 2\exp\left(-\frac{2\log(2T)\|\boldsymbol{x}_t\|_{\boldsymbol{A}_t^{-1}}^2}{\|\boldsymbol{x}_t^\top\boldsymbol{A}_t^{-1}\mathbf{U}_t\|_2^2}\right)$$

$$\leq 2\exp(-2\log(2T))$$

$$\leq \frac{1}{T^2}.$$

The second inequality follows from $\boldsymbol{A}_t = \boldsymbol{I} + \mathbf{U}_t\mathbf{U}_t^\top \succeq \mathbf{U}_t\mathbf{U}_t^\top$. Similarly, we can bound the second term in equation 8 as $\|\boldsymbol{x}_t^\top\boldsymbol{A}_t^{-1}\|_2 = \sqrt{\boldsymbol{x}_t^\top\boldsymbol{A}_t^{-1}\boldsymbol{I}\boldsymbol{A}_t^{-1}\boldsymbol{x}_t} \leq \|\boldsymbol{x}_t\|_{\boldsymbol{A}_t^{-1}}$. Plugging them back in equation 8 yields

$$\left|\widehat{\boldsymbol{\theta}}_t^\top\boldsymbol{x}_t - \boldsymbol{\theta}_*^\top\boldsymbol{x}_t\right| \leq (\sqrt{\log(2T)} + 1)\|\boldsymbol{x}_t\|_{\boldsymbol{A}_t^{-1}}.$$

$\square$

The second part of Lemma 3.1 follows from the elliptical potential lemma that is standard in the bandit literature (Abbasi-Yadkori et al., 2011; Wen et al., 2025):

**Lemma B.2.** *For any given vectors $\{\boldsymbol{x}_\tau\}_{\tau=1}^{t-1}$ in $\mathbb{R}^d$ with $\|\boldsymbol{x}_\tau\|_2 \leq 1$, let the Gram matrix be $\boldsymbol{A}_s = I + \sum_{\tau<s}\boldsymbol{x}_\tau\boldsymbol{x}_\tau^\top$ for every $1 \leq s \leq t$. It holds that*

$$\sum_{\tau<t}\|\boldsymbol{x}_\tau\|_{\boldsymbol{A}_\tau^{-1}}^2 \leq 2d\log\left(1 + \frac{t-1}{d}\right) \leq 2d\log(t).$$

## C NONLINEAR DEMAND ESTIMATION VIA OVER-PARAMETERIZATION

As promised in Section 3.1.2, we build on the idea of NTK (Jacot et al., 2018) and `NeuralUCB` Zhou et al. (2020) to handle nonlinear demand mean using over-parameterized neural networks. For the ease of analysis, we impose a symmetry assumption on the context. Note that for any context $x_t$ with $\|x_t\|_2 \leq 1$, we can apply the transformation $x_t \mapsto [x_t^\top, x_t^\top]^\top/\sqrt{2}$ to satisfy Assumption 6 and incur only an extra constant in the regret.

**Assumption 6.** *The dimension $d$ is even. For any time $t \in [T]$ and any $j \in [d/2]$, it holds that $x_{t,j} = x_{t,j+d/2}$.*

**Initialization** Then we set the network initialization as follows. For each layer $k \in [K-1]$, the weight matrix is set to $\boldsymbol{W}_k = \begin{bmatrix} \boldsymbol{W}_{k,0} & \mathbf{0} \\ \mathbf{0} & \boldsymbol{W}_{k,0} \end{bmatrix}$ where each entry of the initial matrix $\boldsymbol{W}_{k,0}$ is drawn i.i.d. from the Gaussian distribution $\mathcal{N}(0, 4/w)$. The last layer is $\boldsymbol{W}_K = [\boldsymbol{W}_{K,0}^\top, -\boldsymbol{W}_{K,0}^\top]^\top$ where each entry of $\boldsymbol{W}_{K,0}$ is drawn i.i.d. from $\mathcal{N}(0, 2/w)$. The initial parameter is denoted by $\widehat{\boldsymbol{\theta}}_0 \in \mathbb{R}^n$ with dimension $n = (K-2)w^2 + w(d+1)$. By Assumption 6, we always have $f(\boldsymbol{x}; \widehat{\boldsymbol{\theta}}_0) = 0$ for any possible context $\boldsymbol{x}$, which facilitates the convergence analysis. Notably, we sample the initialization $\widehat{\boldsymbol{\theta}}_0$ only *once* and pass this same initialization to the training routine Algorithm 1 every time for the ease of analysis (so our local expansion of $f(\boldsymbol{x}; \widehat{\boldsymbol{\theta}}_t)$ can be around the same initialization $\widehat{\boldsymbol{\theta}}_0$ for every $t$). This needs not be the case in practice.

The NTK matrix is recursively defined as:

**Definition 1** ((Zhou et al., 2020; Jacot et al., 2018)). Let $\{\boldsymbol{x}_t\}_{t\in[T]}$ be the contexts. Define recursively

$$\widetilde{\boldsymbol{H}}_{i,j}^{(1)} = \boldsymbol{\Sigma}_{i,j}^{(1)} = \langle\boldsymbol{x}_i, \boldsymbol{x}_j\rangle, \qquad A_{i,j}^{(k)} = \begin{pmatrix} \boldsymbol{\Sigma}_{i,i}^{(k)} & \boldsymbol{\Sigma}_{i,j}^{(k)} \\ \boldsymbol{\Sigma}_{i,j}^{(k)} & \boldsymbol{\Sigma}_{j,j}^{(k)} \end{pmatrix}$$

$$\boldsymbol{\Sigma}_{i,j}^{(k+1)} = 2\mathbb{E}_{(\mathbf{u},\mathbf{v})\sim\mathcal{N}(0,A_{i,j}^{(k)})}[\langle\sigma(\mathbf{u}), \sigma(\mathbf{v})\rangle]$$

$$\widetilde{\boldsymbol{H}}_{i,j}^{(k+1)} = 2\widetilde{\boldsymbol{H}}_{i,j}^{(k)}2\mathbb{E}_{(\mathbf{u},\mathbf{v})\sim\mathcal{N}(0,A_{i,j}^{(k)})}[\langle\sigma'(\mathbf{u}), \sigma'(\mathbf{v})\rangle] + \boldsymbol{\Sigma}_{i,j}^{(k+1)}$$

Finally, denote the NTK matrix by $\boldsymbol{H} = \frac{1}{2}\Big(\widetilde{\boldsymbol{H}}^{(K)} + \boldsymbol{\Sigma}^{(K)}\Big) \in \mathbb{R}^{T\times T}$.

**Assumption 7** (Non-singular NTK). *Let context $\{\boldsymbol{x}_t\}_{t \in [T]}$ be drawn from the density $f_{\boldsymbol{x}}$. With probability at least $1 - T^{-1}$, $\boldsymbol{H} \succeq \lambda_h \boldsymbol{I}$ for some $\lambda_h > 0$.*

Assumption 7 requires the NTK matrix to be (with high probability under the stochastic context) non-singular. It is a common assumption in the literature on the performance of over-parameterized neural networks in the NTK regime Arora et al. (2019); Cao & Gu (2019); Zhou et al. (2020). If no two contexts lie in parallel, indeed $\boldsymbol{H}$ is full rank and non-singular. As we will see shortly, this lower bound $\lambda_h$ scales inversely with the number of parameters needed to derive our results.

**Definition 2** (Effective dimension). Let $\lambda > 0$ be the regularization parameter used in the training routine Algorithm 1. Define the effective dimension of $\boldsymbol{H}$ to be

$$d_{\boldsymbol{H}} = \frac{\log(\det(\boldsymbol{I} + \boldsymbol{H}/\lambda))}{\log(1 + T/\lambda)}.$$

At a high level, the effective dimension $d_{\boldsymbol{H}}$ captures the intrinsic complexity of the NTK matrix $\boldsymbol{H}$, i.e. the neural network model. Now let $\boldsymbol{f} = [f_*(\boldsymbol{x}_1), \ldots, f_*(\boldsymbol{x}_T)]^\top \in \mathbb{R}^T$ denote the vector of the true demand mean.

### C.1 RESULTS ON APPROXIMATION AND NEAR-INITIALIZATION PROPERTIES

The following lemma shows that, for over-parametrized networks (recall definition in equation 3), the true demand mean $f_*$ can be seen as a linear approximation in terms of the network gradient:

**Lemma C.1** (Lemma 5.1 in Zhou et al. (2020)). *There is a constant $C_0 > 0$ such that for any $w \geq C_0 T^4 K^6 \log(T^2 K/\delta)/\lambda_h^4$, with probability at least $1 - \delta$, we have:*

$$f_*(\boldsymbol{x}_t) = \langle \nabla_2 f(\boldsymbol{x}_t; \widehat{\boldsymbol{\theta}}_0), \boldsymbol{\theta}_* - \widehat{\boldsymbol{\theta}}_0 \rangle,$$

$$\sqrt{w} \|\boldsymbol{\theta}_* - \widehat{\boldsymbol{\theta}}_0\|_2 \leq 2\|\boldsymbol{f}\|_{\boldsymbol{H}^{-1}},$$

*for some $\boldsymbol{\theta}_* \in \mathbb{R}^n$ for every $t \in [T]$. Here $\nabla_2$ denotes the gradient taken with respect to the parameters.*

The proof of this lemma, and those of some of the following lemmas, follows verbatim the proof in Zhou et al. (2020), except that the approximation bound only concerns $T$ contexts instead of $T \cdot \#\{arms\}$ contexts in the bandit setting. Therefore, we refer the readers to the original proof and only present proofs for where there is a notable difference.

Next, we have a few auxiliary lemmas that concern the local properties of the network in the parameter space around initialization. Let the initialization $\boldsymbol{\theta}_0 \in \mathbb{R}^n$ be generated as above and given.

**Lemma C.2** (Lemma 4.1 in Cao & Gu (2019)). *There exists constants $C_1, C_2, C_3 > 0$ such that for any $\delta \in (0,1)$, if $\tau$ satisfies*

$$C_1 w^{-\frac{3}{2}} K^{-\frac{3}{2}} \log(TK^2/\delta)^{\frac{3}{2}} \leq \tau \leq C_2 K^{-6} \log(w)^{-\frac{3}{2}},$$

*then with probability at least $1 - \delta$, for any $\boldsymbol{\theta}, \boldsymbol{\theta}'$ satisfying $\|\boldsymbol{\theta} - \widehat{\boldsymbol{\theta}}_0\|_2 \leq \tau$ and $\|\boldsymbol{\theta}' - \widehat{\boldsymbol{\theta}}_0\|_2 \leq \tau$, we have*

$$|f(\boldsymbol{x}_t; \boldsymbol{\theta}) - f(\boldsymbol{x}_t; \boldsymbol{\theta}') - \langle \nabla_2 f(\boldsymbol{x}_t; \boldsymbol{\theta}'), \boldsymbol{\theta} - \boldsymbol{\theta}' \rangle| \leq C_3 \tau^{\frac{4}{3}} K^3 \sqrt{w \log(w)}$$

*for every $t \in [T]$.*

**Lemma C.3** (Lemma B.3 in Cao & Gu (2019)). *There exists constants $C_1, C_2, C_3 > 0$ such that for any $\delta \in (0,1)$, if $\tau$ satisfies*

$$C_1 w^{-\frac{3}{2}} K^{-\frac{3}{2}} \log(TK^2/\delta)^{\frac{3}{2}} \leq \tau \leq C_2 K^{-6} \log(w)^{-\frac{3}{2}},$$

*then with probability at least $1 - \delta$, for any $\boldsymbol{\theta}$ satisfying $\|\boldsymbol{\theta} - \widehat{\boldsymbol{\theta}}_0\|_2 \leq \tau$, we have*

$$\|\nabla_2 f(\boldsymbol{x}_t; \widehat{\boldsymbol{\theta}}_0)\|_F \leq C_3 \sqrt{Kw}$$

*for every $t \in [T]$.*

**Lemma C.4** (Theorem 5 in Allen-Zhu et al. (2019)). *There exists constants $C_1, C_2, C_3 > 0$ such that for any $\delta \in (0,1)$, if $\tau$ satisfies*

$$C_1 w^{-\frac{3}{2}} K^{-\frac{3}{2}} \max\{\log(T)^{\frac{3}{2}}, \log(w)^{\frac{3}{2}}\} \leq \tau \leq C_2 K^{-\frac{9}{2}} \log(w)^{-3},$$

*then with probability at least $1 - \delta$, for any $\boldsymbol{\theta}$ satisfying $\|\boldsymbol{\theta} - \widehat{\boldsymbol{\theta}}_0\|_2 \leq \tau$, we have*

$$\|\nabla_2 f(\boldsymbol{x}_t; \boldsymbol{\theta}) - \nabla_2 f(\boldsymbol{x}_t; \widehat{\boldsymbol{\theta}}_0)\|_2 \leq C_3 \sqrt{\log(w)} \tau^{\frac{1}{3}} K^3 \|\nabla_2 f(\boldsymbol{x}_t; \widehat{\boldsymbol{\theta}}_0)\|_2$$

*for every $t \in [T]$.*

## C.2 Controlling Post-training Parameters

Recall that we used the routine Algorithm 1 to obtain the parameter $\widehat{\boldsymbol{\theta}}_t$ at each time $t$. While there is a bound for general training epochs $J$, for readability and to derive the desired theoretical guarantees, we only consider $J$ sufficiently large in this work.

**Lemma C.5** (Lemma 5.2 in Zhou et al. (2020))**.** *Suppose* $J = \widetilde{\Omega}(TK/\lambda)$, $w = \widetilde{\Omega}(\max\{K^{24}T^{10}\lambda^{-1}, K^{21}T^{10}\lambda^{-10}\})$, *and* $\eta = O(1/(TKw + w\lambda))$. *Then with probability at least* $1 - T^{-2}/2$, *we have* $\|\widehat{\boldsymbol{\theta}}_t - \widehat{\boldsymbol{\theta}}_0\|_2 \le 2\sqrt{\frac{t}{w\lambda}}$ *and* $\|\boldsymbol{\theta}_* - \widehat{\boldsymbol{\theta}}_t\|_{\boldsymbol{A}_t} \le \frac{\gamma_t}{\sqrt{w}}$ *for every* $t \in [T]$. *Here the normalized Gram matrix of the gradients is defined as*

$$\boldsymbol{A}_t = \frac{1}{w}\sum_{\tau < t}\nabla_2 f(\boldsymbol{x}_\tau; \widehat{\boldsymbol{\theta}}_\tau)\nabla_2 f(\boldsymbol{x}_\tau; \widehat{\boldsymbol{\theta}}_\tau)^\top$$

*and the width coefficients satisfy* $\gamma_t = \widetilde{O}(M\sqrt{d_{\boldsymbol{H}}} + \sqrt{\lambda}\|\boldsymbol{f}\|_{\boldsymbol{H}^{-1}})$.

**Lemma C.6.** *Under the same assumptions as in Lemma C.5, we have*

$$\sum_{t=1}^{T}\min\left\{1, \frac{1}{w}\|\nabla_2 f(\boldsymbol{x}_t; \widehat{\boldsymbol{\theta}}_t)\|^2_{\boldsymbol{A}_t^{-1}}\right\} = \widetilde{O}(d_{\boldsymbol{H}}).$$

While not identical, the proof of Lemma C.6 shares the same lines in that of Lemma 5.4 in Zhou et al. (2020). The dependencies other than $d_{\boldsymbol{H}}$ are negligible under the assumptions that $w$ and $J$ are sufficiently large. For readers familiar with linear contextual bandits, this resembles the elliptical potential lemma (Lemma B.2) in the linear case, as in the NTK regime we have represented our over-parameterized network by a linear form.

We remark that the exponents in the range of the width $w$ have not been optimized, since the theoretical guarantees only serve the purpose of understanding the learning complexity in the NTK regime. In practice, small-scale networks that do not fall into the NTK regime also perform incredibly well, as demonstrated in our experiments in Section 5. This other regime remains much more unexplored in the current deep learning theory.

## C.3 Error Bound for Estimation Oracle

Finally, we give the (formal) restatement of Lemma 3.2 and its proof in this section. In particular, we have the quantity $\widetilde{d} = 1 + d_{\boldsymbol{H}}^2 + d_{\boldsymbol{H}}\|\boldsymbol{f}\|_{\boldsymbol{H}^{-1}}^2$ in Lemma 3.2, with the quantities defined in Definition 1 and Definition 2.

**Lemma C.7** (Restatement of Lemma 3.2)**.** *Suppose* $J = \widetilde{\Omega}(TK/\lambda)$, $\eta = O(1/(TKw + w\lambda))$, *and* $w = \widetilde{\Omega}(\max\{K^{24}T^{10}\lambda^{-1}, K^{21}T^{10}\lambda^{-10}, T^4K^6\lambda_h^{-4}\})$. *Then with probability at least* $1 - T^{-2}$, *we have*

$$\left|f(\boldsymbol{x}_t; \widehat{\boldsymbol{\theta}}_t) - f_*(\boldsymbol{x}_t)\right| \le 2M\left(1 + \sqrt{d_{\boldsymbol{H}}} + \sqrt{\lambda}\|\boldsymbol{f}\|_{\boldsymbol{H}^{-1}}\right)\min\left\{1, w^{-\frac{1}{2}}\|\nabla_2 f(\boldsymbol{x}_t; \widehat{\boldsymbol{\theta}}_t)\|_{\boldsymbol{A}_t^{-1}}\right\} =: \xi_t.$$

*Moreover,* $\sum_{t=1}^{T}\xi_t^2 = \widetilde{O}(1 + d_{\boldsymbol{H}}^2 + d_{\boldsymbol{H}}\|\boldsymbol{f}\|_{\boldsymbol{H}^{-1}}^2)$.

*Proof.* WLOG, suppose the high-probability events in Lemma C.1–C.5 hold. For each $t \in [T]$, the estimation error is bounded as

$$\left|f(\boldsymbol{x}_t; \widehat{\boldsymbol{\theta}}_t) - f_*(\boldsymbol{x}_t)\right|$$

$$\stackrel{(a)}{=} \left|f(\boldsymbol{x}_t; \widehat{\boldsymbol{\theta}}_t) - \langle\nabla_2 f(\boldsymbol{x}_t; \widehat{\boldsymbol{\theta}}_0), \boldsymbol{\theta}_* - \widehat{\boldsymbol{\theta}}_0\rangle\right|$$

$$= \left|f(\boldsymbol{x}_t; \widehat{\boldsymbol{\theta}}_t) - \langle\nabla_2 f(\boldsymbol{x}_t; \widehat{\boldsymbol{\theta}}_t), \widehat{\boldsymbol{\theta}}_t - \widehat{\boldsymbol{\theta}}_0\rangle + \langle\nabla_2 f(\boldsymbol{x}_t; \widehat{\boldsymbol{\theta}}_t), \widehat{\boldsymbol{\theta}}_t - \widehat{\boldsymbol{\theta}}_0\rangle - \langle\nabla_2 f(\boldsymbol{x}_t; \widehat{\boldsymbol{\theta}}_0), \boldsymbol{\theta}_* - \widehat{\boldsymbol{\theta}}_0\rangle\right|$$

$$\le \underbrace{\left|f(\boldsymbol{x}_t; \widehat{\boldsymbol{\theta}}_t) - \langle\nabla_2 f(\boldsymbol{x}_t; \widehat{\boldsymbol{\theta}}_t), \widehat{\boldsymbol{\theta}}_t - \widehat{\boldsymbol{\theta}}_0\rangle\right|}_{(\spadesuit)} + \underbrace{\left|\langle\nabla_2 f(\boldsymbol{x}_t; \widehat{\boldsymbol{\theta}}_t), \widehat{\boldsymbol{\theta}}_t - \widehat{\boldsymbol{\theta}}_0\rangle - \langle\nabla_2 f(\boldsymbol{x}_t; \widehat{\boldsymbol{\theta}}_0), \boldsymbol{\theta}_* - \widehat{\boldsymbol{\theta}}_0\rangle\right|}_{(\diamond)}$$

where (a) applies Lemma C.1. We proceed with term ($\spadesuit$) first. By Lemma C.5, $\|\widehat{\boldsymbol{\theta}}_t - \widehat{\boldsymbol{\theta}}_0\|_2 \leq 2\sqrt{t/(w\lambda)}$. For large enough width $w$ as in the assumption, $\|\widehat{\boldsymbol{\theta}}_t - \widehat{\boldsymbol{\theta}}_0\|_2$ satisfies the assumption in Lemma C.2. Applying Lemma C.2 and the initialization that $f(\boldsymbol{x}; \widehat{\boldsymbol{\theta}}_0) = 0$, we have

$$(\spadesuit) = \left| f(\boldsymbol{x}_t; \widehat{\boldsymbol{\theta}}_t) - f(\boldsymbol{x}; \widehat{\boldsymbol{\theta}}_0) - \langle \nabla_2 f(\boldsymbol{x}_t; \widehat{\boldsymbol{\theta}}_t), \widehat{\boldsymbol{\theta}}_t - \widehat{\boldsymbol{\theta}}_0 \rangle \right| \leq C t^{\frac{2}{3}} K^3 w^{-\frac{1}{6}} \sqrt{\log(w)} \qquad (9)$$

for some constant $C > 0$. To handle term ($\diamondsuit$), observe that by adding and subtracting $\nabla_2 f(\boldsymbol{x}_t; \widehat{\boldsymbol{\theta}}_t)^\top \boldsymbol{\theta}_*$, we have

$$(\diamondsuit) = \left| \langle \nabla_2 f(\boldsymbol{x}_t; \widehat{\boldsymbol{\theta}}_t), \boldsymbol{\theta}_* - \widehat{\boldsymbol{\theta}}_0 \rangle - \langle \nabla_2 f(\boldsymbol{x}_t; \widehat{\boldsymbol{\theta}}_0), \boldsymbol{\theta}_* - \widehat{\boldsymbol{\theta}}_0 \rangle + \langle \nabla_2 f(\boldsymbol{x}_t; \widehat{\boldsymbol{\theta}}_t), \widehat{\boldsymbol{\theta}}_t - \boldsymbol{\theta}_* \rangle \right|$$

$$\leq \underbrace{\left| \langle \nabla_2 f(\boldsymbol{x}_t; \widehat{\boldsymbol{\theta}}_t), \boldsymbol{\theta}_* - \widehat{\boldsymbol{\theta}}_0 \rangle - \langle \nabla_2 f(\boldsymbol{x}_t; \widehat{\boldsymbol{\theta}}_0), \boldsymbol{\theta}_* - \widehat{\boldsymbol{\theta}}_0 \rangle \right|}_{(\heartsuit)} + \underbrace{\left| \langle \nabla_2 f(\boldsymbol{x}_t; \widehat{\boldsymbol{\theta}}_t), \widehat{\boldsymbol{\theta}}_t - \boldsymbol{\theta}_* \rangle \right|}_{(\clubsuit)}.$$

We handle the first term by Cauchy-Schwartz inequality.

$$(\heartsuit) = \left| \langle \nabla_2 f(\boldsymbol{x}_t; \widehat{\boldsymbol{\theta}}_t) - \nabla_2 f(\boldsymbol{x}_t; \widehat{\boldsymbol{\theta}}_0), \boldsymbol{\theta}_* - \widehat{\boldsymbol{\theta}}_0 \rangle \right|$$

$$\leq \|\nabla_2 f(\boldsymbol{x}_t; \widehat{\boldsymbol{\theta}}_t) - \nabla_2 f(\boldsymbol{x}_t; \widehat{\boldsymbol{\theta}}_0)\|_2 \|\boldsymbol{\theta}_* - \widehat{\boldsymbol{\theta}}_0\|_2$$

$$\overset{(b)}{\leq} 2\|\boldsymbol{f}\|_{\boldsymbol{H}^{-1}} w^{-\frac{1}{2}} \|\nabla_2 f(\boldsymbol{x}_t; \widehat{\boldsymbol{\theta}}_t) - \nabla_2 f(\boldsymbol{x}_t; \widehat{\boldsymbol{\theta}}_0)\|_2$$

$$\overset{(c)}{\leq} 4\|\boldsymbol{f}\|_{\boldsymbol{H}^{-1}} w^{-\frac{2}{3}} \sqrt{\log(w)} K^3 t^{\frac{1}{6}} \lambda^{-\frac{1}{6}} \|\nabla_2 f(\boldsymbol{x}_t; \widehat{\boldsymbol{\theta}}_0)\|_2$$

$$\overset{(d)}{\leq} C'\|\boldsymbol{f}\|_{\boldsymbol{H}^{-1}} w^{-\frac{1}{6}} \sqrt{\log(w)} K^{\frac{7}{2}} t^{\frac{1}{6}} \lambda^{-\frac{1}{6}} \qquad (10)$$

where (b) applies Lemma C.1, (c) applies Lemma C.4, and (d) applies Lemma C.3 and gets some constant $C' > 0$. Finally, the last term characterizes the difficulty in learning and does not vanish with $w$ (because the size of the gradient $\nabla_2 f$ also depends on $w$).

$$(\clubsuit) \leq \|\nabla_2 f(\boldsymbol{x}_t; \widehat{\boldsymbol{\theta}}_t)\|_{\boldsymbol{A}_t^{-1}} \|\widehat{\boldsymbol{\theta}}_t - \boldsymbol{\theta}_*\|_{\boldsymbol{A}} \overset{(e)}{\leq} \gamma_t w^{-\frac{1}{2}} \|\nabla_2 f(\boldsymbol{x}_t; \widehat{\boldsymbol{\theta}}_t)\|_{\boldsymbol{A}_t^{-1}}$$

where (e) follows from Lemma C.5. Note naively $\|\boldsymbol{f}\|_{\boldsymbol{H}^{-1}} \leq MT/\lambda_h$. So for the large width $w$ we have in the statement, the other terms in equation 9 and 10 are second-order. We end up with

$$\left| f(\boldsymbol{x}_t; \widehat{\boldsymbol{\theta}}_t) - f_*(\boldsymbol{x}_t) \right| \leq 2\min\left\{ M, \gamma_t w^{-\frac{1}{2}} \|\nabla_2 f(\boldsymbol{x}_t; \widehat{\boldsymbol{\theta}}_t)\|_{\boldsymbol{A}_t^{-1}} \right\}$$

$$\leq 2M\left( 1 + \sqrt{d_{\boldsymbol{H}}} + \sqrt{\lambda}\|\boldsymbol{f}\|_{\boldsymbol{H}^{-1}} \right) \min\left\{ 1, w^{-\frac{1}{2}} \|\nabla_2 f(\boldsymbol{x}_t; \widehat{\boldsymbol{\theta}}_t)\|_{\boldsymbol{A}_t^{-1}} \right\}$$

as desired. The last part of the claim follows from Lemma C.6. $\qquad\square$

# D   LEMMAS ON KERNEL REGRESSION

This section provides proofs for the results in Section 3.3. We will first present results to bound the estimation error when the DM knows $\boldsymbol{z}_t$ precisely. Then in the last section we present results to bound the bias introduced by replacing $\boldsymbol{z}_t$ with $\widehat{\boldsymbol{z}}_t$.

## D.1   KERNEL REGRESSION WITHOUT OBSERVATION ERRORS

**Lemma D.1** (Restatement of Lemma 3.3). *Let the bandwidth be $a_t = t^{-\frac{1}{p+2}} p^{\frac{2}{p+2}}$. Suppose Assumptions 1–3 hold, $t > 1$, and $|\widehat{\epsilon}_\tau - \epsilon_\tau| \leq \xi_\tau$ for every $\tau \in [t-1]$. The NW estimator in equation 5 satisfies: with probability at least $1 - T^{-2}$,*

$$\left| \widehat{Q}(u; \boldsymbol{z}) - Q(u; \boldsymbol{z}) \right| \leq \frac{C_0 \sqrt{\log(T)}}{f_{a_t}(\boldsymbol{z})} \left( L\frac{\overline{\xi}}{t} + t^{-\frac{1}{p+2}} \right)$$

*for every $u \in \mathcal{C}$ and $\boldsymbol{z} \in \mathcal{Z}$, with the constant $C_0$ depending on $K$ and $\overline{f}_{\boldsymbol{z}}$. And $\overline{\xi} = \sum_{\tau < t} \xi_\tau$.*

*Proof.* Let us write $h(u; \boldsymbol{z}) = Q(u; \boldsymbol{z}) f_{\boldsymbol{z}}(\boldsymbol{z})$ for the unconditional CDF. We write the NW estimator as $\widehat{Q}(u; \boldsymbol{z}) = \frac{h_{a_t}(u; \boldsymbol{z})}{f_{a_t}(\boldsymbol{z})}$ with

$$h_{a_t}(u; \boldsymbol{z}) = \frac{1}{t-1} \sum_{\tau \in [t-1]} K\left(\frac{\boldsymbol{z}_\tau - \boldsymbol{z}}{a_t}\right) \mathbb{1}\left[D_\tau - \widehat{D}_\tau \leq u\right] \tag{11}$$

and

$$f_{a_t}(\boldsymbol{z}) = \frac{1}{t-1} \sum_{\tau \in [t-1]} K\left(\frac{\boldsymbol{z}_\tau - \boldsymbol{z}}{a_t}\right). \tag{12}$$

Recall that $\widehat{D}_\tau$ is the oracle-output demand mean estimation, and $\widehat{\epsilon}_\tau = D_\tau - \widehat{D}_\tau$ as set in Algorithm 2. By Lemma D.2 and D.3, we have for every $u \in \frac{1}{T}[T]$ and $\boldsymbol{z} \in \mathcal{Z}$,

$$\max\{|h_{a_t}(u; \boldsymbol{z}) - h(u; \boldsymbol{z})|, |f_{a_t}(\boldsymbol{z}) - f_{\boldsymbol{z}}(\boldsymbol{z})|\}$$
$$\leq C(K, \overline{f}_{\boldsymbol{z}}) \left(a_t + L\overline{\xi}/t + \sqrt{(\log(T) + \log(a_t))p^2/(ta_t^p)}\right)$$

for a constant $C(K, \overline{f}_{\boldsymbol{z}})$ that depends on the kernel $K$ and the bound $\overline{f}_{\boldsymbol{z}}$. Then we choose $a_t \asymp t^{-\frac{1}{p+2}} p^{\frac{2}{p+2}}$ and get

$$\max\{|h_{a_t}(u; \boldsymbol{z}) - h(u; \boldsymbol{z})|, |f_{a_t}(\boldsymbol{z}) - f_{\boldsymbol{z}}(\boldsymbol{z})|\} \leq C(K, \overline{f}_{\boldsymbol{z}})\sqrt{\log(T)}\left(L\overline{\xi}/t + 2t^{-\frac{1}{p+2}} p^{\frac{2}{p+2}}\right)$$
$$\leq 8C(K, \overline{f}_{\boldsymbol{z}})\sqrt{\log(T)}\left(L\overline{\xi}/t + t^{-\frac{1}{p+2}}\right)$$

where the last inequality follows from that $n^{\frac{1}{n}} \leq 2$ for any $n > 0$. Now since

$$\widehat{Q}(u; \boldsymbol{z}) - Q(u; \boldsymbol{z}) = \frac{h_{a_t}(u; \boldsymbol{z})}{f_{a_t}(\boldsymbol{z})} - \frac{h(u; \boldsymbol{z})}{f_{\boldsymbol{z}}(\boldsymbol{z})}$$
$$= \frac{h_{a_t}(u; \boldsymbol{z}) - h(u; \boldsymbol{z})}{f_{a_t}(\boldsymbol{z})} + h(u; \boldsymbol{z})\left(\frac{1}{f_{a_t}(\boldsymbol{z})} - \frac{1}{f_{\boldsymbol{z}}(\boldsymbol{z})}\right), \tag{13}$$

we can obtain the error bound as

$$\left|\widehat{Q}(u; \boldsymbol{z}) - Q(u; \boldsymbol{z})\right| \leq \frac{|h_{a_t}(u; \boldsymbol{z}) - h(u; \boldsymbol{z})|}{f_{a_t}(\boldsymbol{z})} + \frac{h(u; \boldsymbol{z})}{f_{\boldsymbol{z}}(\boldsymbol{z})} \cdot \frac{|f_{a_t}(\boldsymbol{z}) - f_{\boldsymbol{z}}(\boldsymbol{z})|}{f_{a_t}(\boldsymbol{z})}$$
$$= \frac{|h_{a_t}(u; \boldsymbol{z}) - h(u; \boldsymbol{z})|}{f_{a_t}(\boldsymbol{z})} + Q(u; \boldsymbol{z}) \cdot \frac{|f_{a_t}(\boldsymbol{z}) - f_{\boldsymbol{z}}(\boldsymbol{z})|}{f_{a_t}(\boldsymbol{z})}$$
$$= \frac{\sqrt{\log(T)}}{f_{a_t}(\boldsymbol{z})} O\left(L\overline{\xi}/t + t^{-\frac{1}{p+2}}\right).$$

$\square$

Recall that we write $\widehat{Q}(u; \boldsymbol{z}) = \frac{h_{a_t}(u; \boldsymbol{z})}{f_{a_t}(\boldsymbol{z})}$ to approximate $Q(u; \boldsymbol{z}) = \frac{h(u; \boldsymbol{z})}{f_{\boldsymbol{z}}(\boldsymbol{z})}$, as defined in equation 11 and equation 12. In the following, we prove the high-probability error bound for $h_{a_t}$ and $f_{a_t}$ respectively.

**Lemma D.2** (Bias of the estimators). *Suppose Assumptions 1–3 hold, $t > 1$, and $|\widehat{\epsilon}_\tau - \epsilon_\tau| \leq \xi_\tau$ for every $\tau \in [t-1]$. For every $u \in \mathcal{C}$ and $\boldsymbol{z} \in \mathcal{Z}$, we have*

$$|\mathbb{E}[h_{a_t}(u; \boldsymbol{z})] - h(u; \boldsymbol{z})| \leq 2L(1 + \overline{f}_{\boldsymbol{z}})C(K)a_t + 2\overline{f}_{\boldsymbol{z}}L\overline{\xi}/t$$

*and*

$$|\mathbb{E}[f_{a_t}(\boldsymbol{z})] - f_{\boldsymbol{z}}(\boldsymbol{z})| \leq C(K)L_{\boldsymbol{z}}a_t$$

*where the kernel-dependent constant is $C(K) = \int_{\mathbb{R}^p} K(\boldsymbol{y})\|\boldsymbol{y}\| \mathrm{d}\boldsymbol{y}$.*

*Proof.* By straightforward expansion,

$$\mathbb{E}[h_{a_t}(u; \boldsymbol{z})] = \mathbb{E}\left[\frac{1}{t-1}\sum_{\tau \in [t-1]} K_{a_t}(\boldsymbol{z}_\tau - \boldsymbol{z})\mathbb{1}[\widehat{\epsilon}_\tau \leq u | \boldsymbol{z}_\tau]\right]$$

$$= \mathbb{E}\left[\frac{1}{t-1}\sum_{\tau \in [t-1]} K_{a_t}(\boldsymbol{z}_\tau - \boldsymbol{z})Q(u + \epsilon_\tau - \widehat{\epsilon}_\tau; \boldsymbol{z}_\tau)\right]$$

$$\leq \frac{1}{t-1}\sum_{\tau \in [t-1]} \mathbb{E}_{\boldsymbol{z}_\tau \sim f_{\boldsymbol{z}}}[K_{a_t}(\boldsymbol{z}_\tau - \boldsymbol{z})(Q(u; \boldsymbol{z}_\tau) + L|\epsilon_\tau - \widehat{\epsilon}_\tau|)]$$

$$= \mathbb{E}_{\boldsymbol{z}' \sim f_{\boldsymbol{z}}}[K_{a_t}(\boldsymbol{z}' - \boldsymbol{z})Q(u; \boldsymbol{z}')] + \frac{L}{t-1}\sum_{\tau \in [t-1]} \xi_\tau \cdot \mathbb{E}_{\boldsymbol{z}_\tau \sim f_{\boldsymbol{z}}}[K_{a_t}(\boldsymbol{z}_\tau - \boldsymbol{z})]$$

$$\overset{(a)}{\leq} \mathbb{E}_{\boldsymbol{z}' \sim f_{\boldsymbol{z}}}[K_{a_t}(\boldsymbol{z}' - \boldsymbol{z})Q(u; \boldsymbol{z}')] + \frac{\overline{f}_{\boldsymbol{z}}L}{t-1}\overline{\xi}$$

$$= \int_{\mathbb{R}^p} K_{a_t}(\boldsymbol{z}' - \boldsymbol{z})Q(u; \boldsymbol{z}')f_{\boldsymbol{z}}(\boldsymbol{z}')\mathrm{d}\boldsymbol{z}' + \frac{\overline{f}_{\boldsymbol{z}}L}{t-1}\overline{\xi}$$

$$= \int_{\mathbb{R}^p} K_{a_t}(\boldsymbol{z}' - \boldsymbol{z})h(u; \boldsymbol{z}')\mathrm{d}\boldsymbol{z}' + \frac{\overline{f}_{\boldsymbol{z}}L}{t-1}\overline{\xi}$$

$$= \int_{\mathbb{R}^p} K(\boldsymbol{y})h(u; \boldsymbol{z} + a_t\boldsymbol{y})\mathrm{d}\boldsymbol{y} + \frac{\overline{f}_{\boldsymbol{z}}L}{t-1}\overline{\xi}$$

$$\overset{(b)}{\leq} \int_{\mathbb{R}^p} K(\boldsymbol{y})h(u; \boldsymbol{z})\mathrm{d}\boldsymbol{y} + 2L\max\{1, \overline{f}_{\boldsymbol{z}}\}C(K)a_t + \frac{\overline{f}_{\boldsymbol{z}}L}{t-1}\overline{\xi}$$

$$= h(u; \boldsymbol{z}) + 2L\max\{1, \overline{f}_{\boldsymbol{z}}\}C(K)a_t + \frac{\overline{f}_{\boldsymbol{z}}L}{t-1}\overline{\xi}$$

where (a) applies $\mathbb{E}_{\boldsymbol{z}'}[K_{a_t}(\boldsymbol{z}' - \boldsymbol{z})] = \int K_{a_t}(\boldsymbol{z}' - \boldsymbol{z})f_{\boldsymbol{z}}(\boldsymbol{z}')\mathrm{d}\boldsymbol{z}' \leq \overline{f}_{\boldsymbol{z}}$, and (b) uses Assumptions 1 and 3 with constant being $C(K) = \int_{\mathbb{R}^p} K(\boldsymbol{y})\|\boldsymbol{y}\|\mathrm{d}\boldsymbol{y}$. The other direction follows similarly. We use $t > 1$ to write $\frac{1}{t} \leq \frac{1}{t-1} \leq \frac{2}{t}$ in a more convenient way. For the other term,

$$\mathbb{E}[f_{a_t}(\boldsymbol{z})] = \mathbb{E}_{\boldsymbol{z}' \sim f_{\boldsymbol{z}}}[K_{a_t}(\boldsymbol{z}' - \boldsymbol{z})] = \int_{\mathbb{R}^p} K_{a_t}(\boldsymbol{z}' - \boldsymbol{z})f_{\boldsymbol{z}}(\boldsymbol{z}')\mathrm{d}\boldsymbol{z}'$$

$$= \int_{\mathbb{R}^p} K(\boldsymbol{y})f_{\boldsymbol{z}}(\boldsymbol{z} + a_t\boldsymbol{y})\mathrm{d}\boldsymbol{y}$$

$$\leq f_{\boldsymbol{z}}(\boldsymbol{z}) + a_t L_{\boldsymbol{z}}C(K).$$

$\square$

**Lemma D.3** (Deviations of the estimators). *Suppose the bandwidth satisfies $ta_t^p \leq T$. Under Assumptions 1–3 and for $t > 1$, with probability at least $1 - T^{-2}$, we have that*

$$|h_{a_t}(u; \boldsymbol{z}) - \mathbb{E}[h_{a_t}(u; \boldsymbol{z})]| \leq C_2\sqrt{\frac{p^2\log(a_t^{-1}) + \log(4T)}{ta_t^p}}$$

*and*

$$|f_{a_t}(\boldsymbol{z}) - \mathbb{E}[f_{a_t}(\boldsymbol{z})]| \leq C_2'\sqrt{\frac{p^2\log(a_t^{-1}) + \log(4T)}{ta_t^p}}$$

*for every $u \in \mathcal{C}$ and $\boldsymbol{z} \in \mathcal{Z}$, where $C_2$ and $C_2'$ are constants that only depend on $K$ and $\overline{f}_{\boldsymbol{z}}$.*

*Proof.* First, thanks to Assumption 1 and that $\widehat{Q}_t$ is monotone in $u$, it suffices to prove a concentration bound that holds over the fine-enough discretization $u \in \frac{1}{T}[T]$. For $u \notin \frac{1}{T}[T]$, by Lipschitzness of $Q_t$ and monotonicity of $\widehat{Q}_t$, the bound holds with an additional term $O(T^{-1})$ which is subsumed by the dominating terms (as long as $ta_t^p \leq T$).

In the following, we will present the proof for $h_{a_t}(u; \boldsymbol{z})$, since the same argument applies to $f_{a_t}(\boldsymbol{z})$. For any fixed $u \in \frac{1}{T}[T]$ and $\boldsymbol{z} \in \mathcal{Z}$, consider $|h_{a_t}(u; \boldsymbol{z}) - \mathbb{E}[h_{a_t}(u; \boldsymbol{z})]| = \max\{h_{a_t}(u; \boldsymbol{z}) - \mathbb{E}[h_{a_t}(u; \boldsymbol{z})], \mathbb{E}[h_{a_t}(u; \boldsymbol{z})] - h_{a_t}(u; \boldsymbol{z})\}$. We will now bound the first term, and note that the second term can be bounded in the same way.

Let $S^{(i)}$ be $\epsilon_i$-cover of $\mathcal{Z}$ for thresholds $\epsilon_i = 2^{-i}$. For any $\boldsymbol{z}$, let $\boldsymbol{z}^i \in S^{(i)}$ be the covering element of $\boldsymbol{z}$. Denote $Z(u; \boldsymbol{z}) = h_{a_t}(u; \boldsymbol{z}) - \mathbb{E}[h_{a_t}(u; \boldsymbol{z})]$. Then for any $J \in \mathbb{N}$,

$$Z(u; \boldsymbol{z}) = Z(u, \boldsymbol{z}^J) + \sum_{i=J}^{\infty} \big(Z(u; \boldsymbol{z}^{i+1}) - Z(u; \boldsymbol{z}^i)\big). \tag{14}$$

We have the following observations. First,

$$Z(u; \boldsymbol{z}) = \frac{1}{(t-1)a_t^{-p}} \sum_{\tau \in [t-1]} \left( K\left(\frac{\boldsymbol{z}_\tau - \boldsymbol{z}}{a_t}\right) \mathbb{1}[\widehat{\epsilon}_\tau \leq u] - \mathbb{E}_\tau\left[ K\left(\frac{\boldsymbol{z}_\tau - \boldsymbol{z}}{a_t}\right) \mathbb{1}[\widehat{\epsilon}_\tau \leq u]\right] \right)$$

$$=: \frac{1}{(t-1)a_t^{-p}} \sum_{\tau \in [t-1]} A_\tau$$

where each summand satisfies $\mathbb{E}_\tau[A_\tau] = 0$ and $|A_\tau| \leq \|K\|_\infty$. Second, its variance satisfies

$$\mathsf{Var}(A_\tau) \leq \mathbb{E}_\tau\left[ K\left(\frac{\boldsymbol{z}_\tau - \boldsymbol{z}}{a_t}\right)^2 \right] = \int_{R^p} K\left(\frac{\boldsymbol{z}' - \boldsymbol{z}}{a_t}\right)^2 f_{\boldsymbol{z}}(\boldsymbol{z}')\mathrm{d}\boldsymbol{z}'$$

$$= a_t^p \int_{R^p} K(\boldsymbol{y})^2 f_{\boldsymbol{z}}(\boldsymbol{z} + a_t\boldsymbol{y})\mathrm{d}\boldsymbol{y}$$

$$\leq a_t^p \overline{f}_{\boldsymbol{z}} \int_{\mathbb{R}^p} K(\boldsymbol{y})^2 \mathrm{d}\boldsymbol{y} = a_t^p \overline{f}_{\boldsymbol{z}} \|K\|_{L_2}^2.$$

Since $A_\tau$ form a Martingale difference sequence, by Freedman's inequality (Freedman, 1975), we have

$$\mathbb{P}(|Z(u; \boldsymbol{z})| \geq \epsilon) = \mathbb{P}\left( \left| \sum_{\tau \in [t-1]} A_\tau \right| \geq (t-1)a_t^p\epsilon \right) \leq 2\exp\left( -\frac{(t-1)^2 a_t^{2p}\epsilon^2}{2\overline{f}_{\boldsymbol{z}}\|K\|_{L_2}^2 (t-1)a_t^p + \frac{2}{3}\|K\|_\infty (t-1)a_t^p\epsilon} \right)$$

$$\leq 2\exp\left( -2C_1 \frac{(t-1)a_t^p\epsilon^2}{1+\epsilon} \right) \leq 2\exp\big(-C_1(t-1)a_t^p\epsilon^2\big)$$

for constant $C_1 = \frac{8}{\max\{2\overline{f}_{\boldsymbol{z}}\|K\|_{L_2}^2, \frac{2}{3}\|K\|_\infty, 1\}}$, where the last step holds when $\epsilon \leq 1$. By a union bound over $S^{(J)}$,

$$\mathbb{P}\left( \sup_{\boldsymbol{z}} Z(u; \boldsymbol{z}^J) \geq \epsilon \right) \leq \left| S^{(J)} \right| 2\exp\big(-C_1(t-1)a_t^p\epsilon^2\big) \leq 2\exp\big(2p\log(\epsilon_J^{-1}) - C_1(t-1)a_t^p\epsilon^2\big)$$

where we use that the log-covering number of $p$-dimensional unit ball is bounded as $\log\left|S^{(J)}\right| \leq 2p\log(\epsilon_J^{-1})$. Let

$$\epsilon = \sqrt{\frac{2p\log(\epsilon_J^{-1}) + 3\log(4T)}{C_1(t-1)a_t^p}} \asymp \sqrt{\frac{p}{(t-1)a_t^p}}.$$

It holds that

$$\mathbb{P}\left( \sup_{\boldsymbol{z}} Z(u; \boldsymbol{z}^J) \geq \sqrt{\frac{2p\log(\epsilon_J^{-1}) + 3\log(4T)}{C_1(t-1)a_t^p}} \right) \leq \frac{1}{8T^3}. \tag{15}$$

Now, we proceed to handle the differences in the sum in equation 14. For different $\boldsymbol{z}_1$ and $\boldsymbol{z}_2$, we consider

$$Z(u; \boldsymbol{z}_1) - Z(u; \boldsymbol{z}_2) = \frac{1}{(t-1)a_t^p} \sum_{\tau \in [t-1]} B_\tau(u; \boldsymbol{z}_1, \boldsymbol{z}_2)$$

$$:= \frac{1}{(t-1)a_t^p} \sum_{\tau \in [t-1]} K\left(\frac{\boldsymbol{z}_\tau - \boldsymbol{z}_1}{a_t}\right) \mathbb{1}[\widehat{\epsilon}_\tau \leq u] - K\left(\frac{\boldsymbol{z}_\tau - \boldsymbol{z}_2}{a_t}\right) \mathbb{1}[\widehat{\epsilon}_\tau \leq u]$$

$$- \mathbb{E}\left[ K\left(\frac{\boldsymbol{z}_\tau - \boldsymbol{z}_1}{a_t}\right) \mathbb{1}[\widehat{\epsilon}_\tau \leq u] \right] + \mathbb{E}\left[ K\left(\frac{\boldsymbol{z}_\tau - \boldsymbol{z}_2}{a_t}\right) \mathbb{1}[\widehat{\epsilon}_\tau \leq u] \right].$$

Note each term $\mathbb{E}[B_\tau(u; \boldsymbol{z}_1, \boldsymbol{z}_2)] = 0$. Also, let $L_K$ denote the Lipschitz constant of the kernel $K$, and we have

$$
\begin{aligned}
|B_\tau(u; \boldsymbol{z}_1, \boldsymbol{z}_2)| &\leq 2\left|\mathbb{1}[\widehat{\epsilon}_\tau \leq u]\left(K\left(\frac{\boldsymbol{z}_\tau - \boldsymbol{z}_1}{a_t}\right) - K\left(\frac{\boldsymbol{z}_\tau - \boldsymbol{z}_2}{a_t}\right)\right)\right| \\
&\leq 2\left|K\left(\frac{\boldsymbol{z}_\tau - \boldsymbol{z}_1}{a_t}\right) - K\left(\frac{\boldsymbol{z}_\tau - \boldsymbol{z}_2}{a_t}\right)\right| \\
&\leq 2L_K\frac{\|\boldsymbol{z}_1 - \boldsymbol{z}_2\|_2}{a_t}.
\end{aligned}
$$

By Azuma-Hoeffding's inequality (Azuma, 1967; Alon & Spencer, 2016), we have

$$
\begin{aligned}
\mathbb{P}(|Z(u; \boldsymbol{z}_1) - Z(u; \boldsymbol{z}_2)| \geq \epsilon) &= \mathbb{P}\left(\left|\sum_{\tau \in [t-1]} B_\tau(u; \boldsymbol{z}_1, \boldsymbol{z}_2)\right| \geq (t-1)a_t^p\epsilon\right) \\
&\leq 2\exp\left(-\frac{2(t-1)^2 a_t^{2p}\epsilon^2}{2(t-1)L_K\|\boldsymbol{z}_1 - \boldsymbol{z}_2\|_2^2/a_t^2}\right) \\
&= 2\exp\left(-\frac{(t-1)a_t^{2(p+1)}\epsilon^2}{L_K\|\boldsymbol{z}_1 - \boldsymbol{z}_2\|_2^2}\right)
\end{aligned}
$$

Then for $\boldsymbol{z}^i$ and $\boldsymbol{z}^{i+1}$, we have $\|\boldsymbol{z}^i - \boldsymbol{z}^{i+1}\|_2 \leq \epsilon_{i+1} + \epsilon_i = 3\epsilon_i$. Then

$$
\mathbb{P}\left(\left|Z(u; \boldsymbol{z}^i) - Z(u; \boldsymbol{z}^{i+1})\right| \geq \epsilon\right) \leq 2\exp\left(-\frac{(t-1)a_t^{2(p+1)}\epsilon^2}{9L_K\epsilon_i^2}\right).
$$

Again, by union bound over $S^{(i)}$ and $S^{(i+1)}$,

$$
\begin{aligned}
\mathbb{P}\left(\sup_{\boldsymbol{z}}\left|Z(u; \boldsymbol{z}^i) - Z(u; \boldsymbol{z}^{i+1})\right| \geq \epsilon\right) &\leq 2\left|S^{(i)}\right| \cdot \left|S^{(i+1)}\right|\exp\left(-\frac{(t-1)a_t^{2(p+1)}\epsilon^2}{9L_K\epsilon_i^2}\right) \\
&= 2\exp\left(2p\log\frac{1}{\epsilon_i} + 2p\log\frac{2}{\epsilon_i} - \frac{(t-1)a_t^{2(p+1)}\epsilon^2}{9L_K\epsilon_i^2}\right) \\
&\leq 2\exp\left(6pi\log 2 - \frac{(t-1)a_t^{2(p+1)}\epsilon^2}{9L_K\epsilon_i^2}\right).
\end{aligned}
$$

Define a target threshold for each level $i \geq J$:

$$
\epsilon(i) = \frac{3\sqrt{L_K}\epsilon_i}{\sqrt{t-1}a_t^{p+1}}\sqrt{7pi\log 2 + 3\log(4T)} = \frac{3\sqrt{L_K}2^{-i}}{\sqrt{t-1}a_t^{p+1}}\sqrt{7pi\log 2 + 3\log(4T)}.
$$

We first note that

$$
\begin{aligned}
\sum_{i=J}^{\infty}\epsilon(i) &= \frac{3\sqrt{L_K}}{\sqrt{t-1}a_t^{p+1}}\sum_{i=J}^{\infty}\frac{\sqrt{7pi + 3\log(4T)}}{2^i} \leq \frac{3\sqrt{L_K}}{\sqrt{t-1}a_t^{p+1}}\sum_{i=J}^{\infty}\left(\frac{i}{2^i}\sqrt{\frac{7p}{J}} + \frac{\sqrt{3\log(4T)}}{2^i}\right) \\
&\leq \frac{3\sqrt{L_K}}{\sqrt{t-1}a_t^{p+1}}\left(\frac{\sqrt{7Jp} + \sqrt{3\log(4T)}}{2^{J-2}}\right).
\end{aligned}
$$

Then

$$\mathbb{P}\left(\sup_{\boldsymbol{z}}\left|Z(u;\boldsymbol{z}) - Z(u;\boldsymbol{z}^J)\right| \geq \frac{3\sqrt{L_K}}{\sqrt{t-1}a_t^{p+1}}\left(\frac{\sqrt{7Jp} + \sqrt{3\log(4T)}}{2^{J-2}}\right)\right)$$

$$\leq \mathbb{P}\left(\sup_{\boldsymbol{z}}\left|Z(u;\boldsymbol{z}) - Z(u;\boldsymbol{z}^J)\right| \geq \sum_{i=J}^{\infty}\epsilon(i)\right)$$

$$\leq \mathbb{P}\left(\sum_{i=J}^{\infty}\sup_{\boldsymbol{z}}\left|Z(u;\boldsymbol{z}^{i+1}) - Z(u;\boldsymbol{z}^i)\right| \geq \sum_{i=M}^{\infty}\epsilon(i)\right)$$

$$\leq \sum_{i=J}^{\infty}\mathbb{P}\left(\sup_{\boldsymbol{z}}\left|Z(u;\boldsymbol{z}^{i+1}) - Z(u;\boldsymbol{z}^i)\right| \geq \epsilon(i)\right)$$

$$\leq \sum_{i=J}^{\infty}2^{-pi}\frac{1}{8T^3} \leq \frac{1}{8T^3}\sum_{i=J}^{\infty}2^{-i} \leq \frac{1}{8T^3}. \tag{16}$$

Combining equation 16 and equation 15, we have

$$\frac{1}{4T^3} \geq \mathbb{P}\left(\sup_{\boldsymbol{z}}\left|Z(u;\boldsymbol{z}) - Z(u;\boldsymbol{z}^J)\right| \geq \frac{3\sqrt{L_K}}{\sqrt{t-1}a_t^{p+1}}\left(\frac{\sqrt{7Jp} + \sqrt{3\log(4T)}}{2^{J-2}}\right)\right)$$

$$+ \mathbb{P}\left(\sup_{\boldsymbol{z}} Z(u;\boldsymbol{z}^J) \geq \sqrt{\frac{2p\log\left(\epsilon_J^{-1}\right) + 3\log(4T)}{C_1(t-1)a_t^p}}\right)$$

$$\geq \mathbb{P}\left(\sup_{\boldsymbol{z}} Z(u;\boldsymbol{z}) \geq \sqrt{\frac{2p\log\left(\epsilon_J^{-1}\right) + 3\log(4T)}{C_1(t-1)a_t^p}} + \frac{3\sqrt{L_K}}{\sqrt{t-1}a_t^{p+1}}\left(\frac{\sqrt{7Jp} + \sqrt{3\log(4T)}}{2^{J-2}}\right)\right)$$

$$\geq \mathbb{P}\left(\sup_{\boldsymbol{z}} Z(u;\boldsymbol{z}) \geq \sqrt{\frac{2pJ + 3\log(4T)}{C_1(t-1)a_t^p}} + \frac{3\sqrt{L_K}}{\sqrt{t-1}a_t^{p+1}}\left(\frac{\sqrt{7Jp} + \sqrt{3\log(4T)}}{2^{J-2}}\right)\right).$$

To balance the terms, we choose $J = \log\left(a_t^{-\frac{p+2}{2}}\right) = \frac{p+2}{2}\log\left(a_t^{-1}\right)$. Then there is another constant $C_2$ that depends on $L_K$, $\overline{f}_{\boldsymbol{z}}$, $\|K\|_\infty$, and $\|K\|_{L_2}^2$, such that

$$\frac{1}{4T^3} \geq \mathbb{P}\left(\sup_{\boldsymbol{z}} Z(u;\boldsymbol{z}) \geq C_2\sqrt{\frac{p^2\log(a_t^{-1}) + \log(4T)}{(t-1)a_t^p}}\right). \tag{17}$$

Note here we have also used $\sqrt{a+b} \leq \sqrt{a} + \sqrt{b} \leq 2\sqrt{a+b}$ for $a, b > 0$. By the same argument, we also bound

$$\frac{1}{4T^3} \geq \mathbb{P}\left(\sup_{\boldsymbol{z}} -Z(u;\boldsymbol{z}) \geq C_2\sqrt{\frac{p^2\log(a_t^{-1}) + \log(4T)}{(t-1)a_t^p}}\right). \tag{18}$$

Taking a union bound over $u \in \frac{1}{T}[T]$ and equation 17 and equation 18 yields the desired bound for $|h_{a_t}(u;\boldsymbol{z}) - \mathbb{E}[h_{a_t}(u;\boldsymbol{z})]|$. The second part of the claim follows the same proof. Finally, note $t - 1 \geq \frac{t}{2}$ for $t \geq 2$. $\qquad\square$

## D.2 KERNEL REGRESSION UNDER BENIGN NOISE

The following lemma bounds the error $\left|\widehat{Q}(u;\boldsymbol{z}) - Q(u;\boldsymbol{z})\right|$ at each time $t$, which proves Lemma 3.5 when the DM observes the precise features $\boldsymbol{z}_t$. To obtain Lemma 3.5 with general observation errors, one simply apply Lemma D.4 and Lemma D.6 in the next section with the chosen bandwidth.

**Lemma D.4** (Lemma 3.5 with precise features)**.** *Suppose Assumptions 1–3 and 5 holds, $t > 1$, and $|\widehat{\epsilon}_\tau - \epsilon_\tau| \leq \xi_\tau$ for every $\tau \in [t-1]$. Let the bandwidth be $a_t = c_\kappa\left(\frac{c_{FT}}{\log(T)}\right)^{\frac{1}{\omega}}$. Then there exists*

*a kernel $K$, defined as in equation 19, such that the NW estimator in equation 5 satisfies: with probability at least $1 - T^{-2}$,*

$$\left|\widehat{Q}(u; \boldsymbol{z}) - Q(u; \boldsymbol{z})\right| \leq \frac{\gamma'}{f_{a_t}(\boldsymbol{z})}\left(L\frac{\overline{\xi}}{t} + \frac{p}{\sqrt{t}}\right)$$

*for every $u \in \mathcal{C}$ and $\boldsymbol{z} \in \mathcal{Z}$, with $\gamma' = O\left(\log(T)^{\frac{p}{\omega}} \log\log(T)^{\frac{1}{2}}\right)$.*

*Proof.* The proof is the same as that of Lemma D.1, except for replacing Lemma D.2 by D.5. □

The next result improves on Lemma D.2 under the additional regularity condition in Assumption 5. In particular, we consider a kernel defined as follows (Fan et al., 2024):

$$K(\boldsymbol{y}) = \mathcal{T}^{-1}[\kappa](\boldsymbol{y}) = \frac{1}{(2\pi)^p}\int_{\mathbb{R}^p}\kappa(\boldsymbol{z})\exp(i\boldsymbol{y}^\top\boldsymbol{z})\mathrm{d}\boldsymbol{z} \tag{19}$$

which is the inverse Fourier Transform of some regular function $\kappa : \mathbb{R}^p \to \mathbb{R}$ that satisfies

$$\kappa(\boldsymbol{z}) = \begin{cases} 1, & \|\boldsymbol{z}\|_2 \leq c_\kappa, \\ g_\kappa(\|\boldsymbol{z}\|_2), & \|\boldsymbol{z}\|_2 > c_\kappa \end{cases}$$

for some constant $c_\kappa > 0$ and function $g_\kappa \in C^0 \cup L_2$ with $\|g_\kappa\|_\infty \leq \overline{g}_\kappa$ and $g_\kappa(c_\kappa) = 1$ (to make it continuous).

**Lemma D.5.** *Suppose Assumptions 1–3 and 5 hold, $t > 1$, and $|\widehat{\epsilon}_\tau - \epsilon_\tau| \leq \xi_\tau$ for every $\tau \in [t-1]$. Recall we denote $\overline{\xi} = \sum_{\tau < t}\xi_\tau$. For every $u \in \mathcal{C}$ and $\boldsymbol{z} \in \mathcal{Z}$, we have*

$$|\mathbb{E}[h_{a_t}(\boldsymbol{z})] - h(u; \boldsymbol{z})| \leq \frac{\overline{C}_{FT}}{\sqrt{T}} + 2\overline{f}_{\boldsymbol{z}}L\overline{\xi}/t$$

*and*

$$|\mathbb{E}[f_{a_t}(\boldsymbol{z})] - f_{\boldsymbol{z}}(\boldsymbol{z})| \leq \frac{\overline{C}_{FT}}{\sqrt{T}}$$

*with bandwidth $a_t = c_\kappa\left(\frac{c_{FT}}{\log(T)}\right)^{\frac{1}{\omega}}$, where the constant is*

$$\overline{C}_{FT} = \frac{\overline{g}_\kappa + 1}{(2\pi)^p}\int_{\mathbb{R}^p} C_{FT}\exp(-c_{FT}\|\boldsymbol{y}\|_2^\omega)\mathrm{d}\boldsymbol{y}.$$

*Proof.* We will prove this argument for $f_{\boldsymbol{z}}$, and the proof for $h(u; \boldsymbol{z})$ follows similarly as in Lemma D.2. Let us denote $\phi = \mathcal{T}[f_{\boldsymbol{z}}]$. Recall from Lemma D.2 that we have

$$\begin{aligned} \mathbb{E}[f_{a_t}(\boldsymbol{z})] - f_{\boldsymbol{z}}(\boldsymbol{z}) &= \int_{\mathbb{R}^p} K_{a_t}(\boldsymbol{z}' - \boldsymbol{z})f_{\boldsymbol{z}}(\boldsymbol{z}')\mathrm{d}\boldsymbol{z}' - f_{\boldsymbol{z}}(\boldsymbol{z}) \\ &\overset{(a)}{=} \mathcal{T}^{-1}\circ(\mathcal{T}[f_{\boldsymbol{z}}(\cdot)]\mathcal{T}[K_{a_t}(-\cdot)] - \mathcal{T}[f_{\boldsymbol{z}}(\cdot)])(\boldsymbol{z}) \\ &= \mathcal{T}^{-1}[\phi(\cdot)(\kappa(-a_t\cdot) - 1)](\boldsymbol{z}) \\ &= \frac{1}{(2\pi)^p}\int_{\mathbb{R}^p}\phi(\boldsymbol{y})(\kappa(-a_t\boldsymbol{y}) - 1)\exp(i\boldsymbol{z}^\top\boldsymbol{y})\mathrm{d}\boldsymbol{y} \end{aligned}$$

where (a) uses the convolution theorem. Then by our choice of $\kappa$, we have

$$
\begin{aligned}
|\mathbb{E}[f_{a_t}(\boldsymbol{z})] - f_{\boldsymbol{z}}(\boldsymbol{z})| &\leq \frac{1}{(2\pi)^p} \int_{\mathbb{R}^p} |\phi(\boldsymbol{y})(\kappa(-a_t\boldsymbol{y}) - 1) \exp(i\boldsymbol{z}^\top\boldsymbol{y})| \mathrm{d}\boldsymbol{y} \\
&\leq \frac{1}{(2\pi)^p} \int_{\mathbb{R}^p} |\phi(\boldsymbol{y})||\kappa(-a_t\boldsymbol{y}) - 1|\mathrm{d}\boldsymbol{y} \\
&= \frac{1}{(2\pi)^p} \int_{\boldsymbol{y}:\|\boldsymbol{y}\|_2 > \frac{c_\kappa}{a_t}} |\phi(\boldsymbol{y})||\kappa(-a_t\boldsymbol{y}) - 1|\mathrm{d}\boldsymbol{y} \\
&\leq \frac{\overline{g}_\kappa + 1}{(2\pi)^p} \int_{\boldsymbol{y}:\|\boldsymbol{y}\|_2 > \frac{c_\kappa}{a_t}} |\phi(\boldsymbol{y})|\mathrm{d}\boldsymbol{y} \\
&\overset{(b)}{\leq} \frac{\overline{g}_\kappa + 1}{(2\pi)^p} \int_{\boldsymbol{y}:\|\boldsymbol{y}\|_2 > \frac{c_\kappa}{a_t}} C_{FT} \exp(-c_{FT}\|\boldsymbol{y}\|_2^\omega)\mathrm{d}\boldsymbol{y} \\
&\leq \frac{\overline{g}_\kappa + 1}{(2\pi)^p} \int_{\mathbb{R}^p} C_{FT} \exp\left(-c_{FT}\left(\|\boldsymbol{y}\|_2 + \frac{c_\kappa}{a_t}\right)^\omega\right)\mathrm{d}\boldsymbol{y} \\
&\leq \frac{\overline{g}_\kappa + 1}{(2\pi)^p} \int_{\mathbb{R}^p} C_{FT} \exp\left(-c_{FT}\left(\|\boldsymbol{y}\|_2^\omega + \left(\frac{c_\kappa}{a_t}\right)^\omega\right)\right)\mathrm{d}\boldsymbol{y}
\end{aligned}
$$

where (b) is from Assumption 5. Choose $a_t = c_\kappa\left(\frac{c_{FT}}{\log(T)}\right)^{\frac{1}{\omega}}$ and we arrive at

$$
|\mathbb{E}[f_{a_t}(\boldsymbol{z})] - f_{\boldsymbol{z}}(\boldsymbol{z})| \leq \frac{\overline{C}_{FT}}{\sqrt{T}}
$$

with constant $\overline{C}_{FT} = \frac{\overline{g}_\kappa + 1}{(2\pi)^p} \int_{\mathbb{R}^p} C_{FT} \exp(-c_{FT}\|\boldsymbol{y}\|_2^\omega)\mathrm{d}\boldsymbol{y}$. $\qquad\square$

### D.3 KERNEL REGRESSION WITH FEATURE ERRORS

Recall that the DM applies kernel regression to the potentially inaccurate features $\{\widehat{\boldsymbol{z}}_\tau\}_{\tau<t}$ and target $\widehat{\boldsymbol{z}}$, with the guarantee that $\|\widehat{\boldsymbol{z}}_\tau - \boldsymbol{z}_\tau\|_2 \leq \delta_\tau$ and $\|\widehat{\boldsymbol{z}} - \boldsymbol{z}\|_2 \leq \delta_t$. Also recall that $\delta_\tau$ is assumed to be $\mathcal{F}_\tau$-measurable. And the NW estimator in equation 6 is built as

$$
\widetilde{Q}_t(u; \widehat{\boldsymbol{z}}) = \frac{\frac{1}{t-1}\sum_{\tau=1}^{t-1} K_{a_t}(\widehat{\boldsymbol{z}}_\tau - \widehat{\boldsymbol{z}})\mathbb{1}[\widehat{\epsilon}_\tau \leq u]}{\frac{1}{t-1}\sum_{\tau=1}^{t-1} K_{a_t}(\widehat{\boldsymbol{z}}_\tau - \widehat{\boldsymbol{z}})} =: \frac{\widetilde{h}_{a_t}(u; \boldsymbol{z})}{\widetilde{f}_{a_t}(\boldsymbol{z})}.
$$

Note the functions $\widetilde{h}_{a_t}$ and $\widetilde{f}_{a_t}$ depend on $\boldsymbol{z}$ through the erroneous observation $\widehat{\boldsymbol{z}}$. It now suffices to bound the difference $\left|\widetilde{Q}_t(u; \widehat{\boldsymbol{z}}) - \widehat{Q}_t(u; \boldsymbol{z})\right|$, given Lemma 3.3. Towards this goal and similar to equation 13, observe that

$$
\begin{aligned}
\widetilde{Q}_t(u; \widehat{\boldsymbol{z}}) - \widehat{Q}_t(u; \boldsymbol{z}) &= \frac{\widetilde{h}_{a_t}(u; \boldsymbol{z})}{\widetilde{f}_{a_t}(\boldsymbol{z})} - \frac{h_{a_t}(u; \boldsymbol{z})}{f_{a_t}(\boldsymbol{z})} \\
&= \frac{\widetilde{h}_{a_t}(u; \boldsymbol{z}) - h_{a_t}(u; \boldsymbol{z})}{f_{a_t}(\boldsymbol{z})} + \widetilde{h}_{a_t}(u; \boldsymbol{z})\left(\frac{1}{\widetilde{f}_{a_t}(\boldsymbol{z})} - \frac{1}{f_{a_t}(\boldsymbol{z})}\right) \\
&= \frac{\widetilde{h}_{a_t}(u; \boldsymbol{z}) - h_{a_t}(u; \boldsymbol{z})}{f_{a_t}(\boldsymbol{z})} + \frac{\widetilde{h}_{a_t}(u; \boldsymbol{z})}{\widetilde{f}_{a_t}(\boldsymbol{z})} \cdot \frac{f_{a_t}(\boldsymbol{z}) - \widetilde{f}_{a_t}(\boldsymbol{z})}{f_{a_t}(\boldsymbol{z})} \\
&= \frac{\widetilde{h}_{a_t}(u; \boldsymbol{z}) - h_{a_t}(u; \boldsymbol{z})}{f_{a_t}(\boldsymbol{z})} + \widetilde{Q}_t(u; \widehat{\boldsymbol{z}})\frac{f_{a_t}(\boldsymbol{z}) - \widetilde{f}_{a_t}(\boldsymbol{z})}{f_{a_t}(\boldsymbol{z})}.
\end{aligned}
$$

Since $\widetilde{Q}_t(u; \widehat{\boldsymbol{z}}) \in [0, 1]$, we have

$$
\left|\widetilde{Q}_t(u; \widehat{\boldsymbol{z}}) - \widehat{Q}_t(u; \boldsymbol{z})\right| \leq \frac{\left|\widetilde{h}_{a_t}(u; \boldsymbol{z}) - h_{a_t}(u; \boldsymbol{z})\right| + \left|\widetilde{f}_{a_t}(\boldsymbol{z}) - f_{a_t}(\boldsymbol{z})\right|}{f_{a_t}(\boldsymbol{z})}. \tag{20}
$$

Let us first consider the term $\left|\widetilde{h}_{a_t}(u; \boldsymbol{z}) - h_{a_t}(u; \boldsymbol{z})\right|$. It holds that

$$\left|\widetilde{h}_{a_t}(u; \boldsymbol{z}) - h_{a_t}(u; \boldsymbol{z})\right| \leq \left|\mathbb{E}\left[\widetilde{h}_{a_t}(u; \boldsymbol{z}) - h_{a_t}(u; \boldsymbol{z})\right]\right| + \left|\widetilde{h}_{a_t}(u; \boldsymbol{z}) - \mathbb{E}\left[\widetilde{h}_{a_t}(u; \boldsymbol{z})\right]\right|$$
$$+ \left|h_{a_t}(u; \boldsymbol{z}) - \mathbb{E}[h_{a_t}(u; \boldsymbol{z})]\right| \tag{21}$$

where the last term is readily handled by Lemma D.3. We remark that the deviation term $\left|\widetilde{h}_{a_t}(u; \boldsymbol{z}) - \mathbb{E}\left[\widetilde{h}_{a_t}(u; \boldsymbol{z})\right]\right|$ can be bounded following almost the same lines of the proof of Lemma D.3, and is hence omitted. This holds thanks to the fact that the errors $\delta_\tau$ is $\mathcal{F}_\tau$-measurable and thus the source of the variance in $\widetilde{h}_{a_t}$ is the same as that in $h_{a_t}$. Then the difference in expectation is as follows:

$$\left|\mathbb{E}\left[\widetilde{h}_{a_t}(u; \boldsymbol{z}) - h_{a_t}(u; \boldsymbol{z})\right]\right|$$
$$\leq \frac{1}{t-1} \sum_{\tau < t} \left|\mathbb{E}_\tau[(K_{a_t}(\widehat{\boldsymbol{z}}_\tau - \widehat{\boldsymbol{z}}) - K_{a_t}(\boldsymbol{z}_\tau - \boldsymbol{z}))Q(u + \epsilon_\tau - \widehat{\epsilon}_\tau; \boldsymbol{z}_\tau)]\right|$$
$$= \frac{1}{t-1} \sum_{\tau < t} \left|\int_{\mathbb{R}^p} (K_{a_t}(\widehat{\boldsymbol{z}}_\tau - \widehat{\boldsymbol{z}}) - K_{a_t}(\boldsymbol{z}_\tau - \boldsymbol{z}))Q(u + \epsilon_\tau - \widehat{\epsilon}_\tau; \boldsymbol{z}_\tau)f_{\boldsymbol{z}}(\boldsymbol{z}_\tau)\mathrm{d}\boldsymbol{z}_\tau\right|$$
$$= \frac{1}{t-1} \sum_{\tau < t} \left|\int_{\mathbb{R}^p} (K_{a_t}(\widehat{\boldsymbol{z}}_\tau - \widehat{\boldsymbol{z}}) - K_{a_t}(\boldsymbol{z}_\tau - \boldsymbol{z}))Q(u + \epsilon_\tau - \widehat{\epsilon}_\tau; \boldsymbol{z}_\tau)f_{\boldsymbol{z}}(\boldsymbol{z}_\tau)\mathrm{d}\boldsymbol{z}_\tau\right|. \tag{22}$$

By Assumption 1, $Q(u + \epsilon_\tau - \widehat{\epsilon}_\tau; \boldsymbol{z}_\tau) \leq Q(u; \boldsymbol{z}_\tau) + L \cdot |\epsilon_\tau - \widehat{\epsilon}_\tau| \leq Q(u; \boldsymbol{z}_\tau) + L\xi_\tau$ contributes an error that depends on the estimation of $\theta_*$. Then

$$\int_{\mathbb{R}^p} (K_{a_t}(\widehat{\boldsymbol{z}}_\tau - \widehat{\boldsymbol{z}}) - K_{a_t}(\boldsymbol{z}_\tau - \boldsymbol{z}))Q(u + \epsilon_\tau - \widehat{\epsilon}_\tau; \boldsymbol{z}_\tau)f_{\boldsymbol{z}}(\boldsymbol{z}_\tau)\mathrm{d}\boldsymbol{z}_\tau$$
$$\leq \int_{\mathbb{R}^p} (K_{a_t}(\widehat{\boldsymbol{z}}_\tau - \widehat{\boldsymbol{z}}) - K_{a_t}(\boldsymbol{z}_\tau - \boldsymbol{z}))Q(u; \boldsymbol{z}_\tau)f_{\boldsymbol{z}}(\boldsymbol{z}_\tau)\mathrm{d}\boldsymbol{z}_\tau$$
$$+ L\xi_\tau \int_{\mathbb{R}^p} (K_{a_t}(\widehat{\boldsymbol{z}}_\tau - \widehat{\boldsymbol{z}}) - K_{a_t}(\boldsymbol{z}_\tau - \boldsymbol{z}))f_{\boldsymbol{z}}(\boldsymbol{z}_\tau)\mathrm{d}\boldsymbol{z}_\tau$$
$$= \int_{\mathbb{R}^p} (K_{a_t}(\widehat{\boldsymbol{z}}_\tau - \widehat{\boldsymbol{z}}) - K_{a_t}(\boldsymbol{z}_\tau - \boldsymbol{z}))h(u; \boldsymbol{z}_\tau)\mathrm{d}\boldsymbol{z}_\tau$$
$$+ L\xi_\tau \int_{\mathbb{R}^p} (K_{a_t}(\widehat{\boldsymbol{z}}_\tau - \widehat{\boldsymbol{z}}) - K_{a_t}(\boldsymbol{z}_\tau - \boldsymbol{z}))f_{\boldsymbol{z}}(\boldsymbol{z}_\tau)\mathrm{d}\boldsymbol{z}_\tau$$
$$= \underbrace{\int_{\mathbb{R}^p} \left(K\left(\boldsymbol{y} + \frac{(\widehat{\boldsymbol{z}}_\tau - \boldsymbol{z}_\tau) + (\widehat{\boldsymbol{z}} - \boldsymbol{z})}{a_t}\right) - K(\boldsymbol{y})\right)h(u; \boldsymbol{z}_\tau + a_t\boldsymbol{y})\mathrm{d}\boldsymbol{y}}_{(A)}$$
$$+ \underbrace{L\xi_\tau \int_{\mathbb{R}^p} \left(K\left(\boldsymbol{y} + \frac{(\widehat{\boldsymbol{z}}_\tau - \boldsymbol{z}_\tau) + (\widehat{\boldsymbol{z}} - \boldsymbol{z})}{a_t}\right) - K(\boldsymbol{y})\right)f_{\boldsymbol{z}}(\boldsymbol{z}_\tau + a_t\boldsymbol{y})\mathrm{d}\boldsymbol{y}}_{(B)}.$$

Let $\boldsymbol{v}_\tau := \frac{(\widehat{\boldsymbol{z}}_\tau - \boldsymbol{z}_\tau) + (\widehat{\boldsymbol{z}} - \boldsymbol{z})}{a_t}$ marks the offset for the ease of notation. The first term (A) proceeds as follows:

$$(A) = \int_{\mathbb{R}^p} (K(\boldsymbol{y} + \boldsymbol{v}_\tau) - K(\boldsymbol{y}))h(u; \boldsymbol{z}_\tau + a_t\boldsymbol{y})\mathrm{d}\boldsymbol{y}$$

$$\overset{(a)}{\leq} \underbrace{\int_{\mathbb{R}^p} (K(\boldsymbol{y} + \boldsymbol{v}_\tau) - K(\boldsymbol{y}))h(u; \boldsymbol{z}_\tau)\mathrm{d}\boldsymbol{y}}_{=0 \text{ as } h(u; \boldsymbol{z}_\tau) \text{ is irrelevant to } \boldsymbol{y}} + a_t^p \overline{f}_{\boldsymbol{z}} L \int_{\mathbb{R}^p} (K(\boldsymbol{y} + \boldsymbol{v}_\tau) - K(\boldsymbol{y}))\|\boldsymbol{y}\|_2 \mathrm{d}\boldsymbol{y}$$

$$= a_t^p \overline{f}_{\boldsymbol{z}} L \left( \int_{\mathbb{R}^p} K(\boldsymbol{y} + \boldsymbol{v}_\tau)\|\boldsymbol{y}\|_2 \mathrm{d}\boldsymbol{y} - \int_{\mathbb{R}^p} K(\boldsymbol{y})\|\boldsymbol{y}\| \mathrm{d}\boldsymbol{y} \right)$$

$$\overset{(b)}{\leq} a_t^p \overline{f}_{\boldsymbol{z}} L \left( \int_{\mathbb{R}^p} K(\boldsymbol{y}')\|\boldsymbol{y}'\|_2 \mathrm{d}\boldsymbol{y}' + \int_{\mathbb{R}^p} K(\boldsymbol{y}')\|\boldsymbol{v}_\tau\|_2 \mathrm{d}\boldsymbol{y}' - \int_{\mathbb{R}^p} K(\boldsymbol{y})\|\boldsymbol{y}\|_2 \mathrm{d}\boldsymbol{y} \right)$$

$$= a_t^p \overline{f}_{\boldsymbol{z}} L \|\boldsymbol{v}_\tau\|_2$$

$$\leq \overline{f}_{\boldsymbol{z}} L (\|\widehat{\boldsymbol{z}}_\tau - \boldsymbol{z}_\tau\|_2 + \|\widehat{\boldsymbol{z}} - \boldsymbol{z}\|_2) \leq \overline{f}_{\boldsymbol{z}} L (\delta_\tau + \delta_t) \tag{23}$$

where (a) uses the Lipschitz constant $\overline{f}_{\boldsymbol{z}} L$ of the conditional probability $h(u; \boldsymbol{z}) = Q(u; \boldsymbol{z})f_{\boldsymbol{z}}(\boldsymbol{z})$ from Assumption 1 and 3, and (b) applies a change of variable $\boldsymbol{y}' = \boldsymbol{y} + \boldsymbol{v}_\tau$.

Similarly, the other term is

$$(B) \leq L_{\boldsymbol{z}} L \xi_\tau (\delta_\tau + \delta_t). \tag{24}$$

Combining equation 23, 24, and equation 22, we obtain

$$\left| \mathbb{E}\left[ \widetilde{h}_{a_t}(u; \boldsymbol{z}) - h_{a_t}(u; \boldsymbol{z}) \right] \right| \leq \frac{L}{t-1} \sum_{\tau < t} (\overline{f}_{\boldsymbol{z}} + L_{\boldsymbol{z}} \xi_\tau)(\delta_\tau + \delta_t).$$

Substituting this back in equation 21 and applying Lemma D.3 on the deviation terms leads to, with probability at least $1 - 2T^{-2}$,

$$\left| \widetilde{h}_{a_t}(u; \boldsymbol{z}) - h_{a_t}(u; \boldsymbol{z}) \right| \leq \frac{\widetilde{C}_2}{t-1} \sum_{\tau < t} (1 + \xi_\tau)(\delta_\tau + \delta_t) + 2C_2 \sqrt{\frac{p^2 \log(a_t^{-1}) + \log(4T)}{t a_t^p}} \tag{25}$$

where $\widetilde{C}_2 = L \max\{\overline{f}_{\boldsymbol{z}}, L_{\boldsymbol{z}}\}$ depends on the Lipschitz constants and density bound, and $C_2$ is the constant in Lemma D.3 that depends on both $K$ and $\overline{f}_{\boldsymbol{z}}$.

Applying the same argument to $\widetilde{f}_{a_t}$ gives the bound with probability at least $1 - 2T^{-2}$,

$$\left| \widetilde{f}_{a_t}(\boldsymbol{z}) - f_{a_t}(\boldsymbol{z}) \right| \leq \frac{L_{\boldsymbol{z}} L}{t-1} \sum_{\tau < t} (1 + \xi_\tau)(\delta_\tau + \delta_t) + 2C_2' \sqrt{\frac{p^2 \log(a_t^{-1}) + \log(4T)}{t a_t^p}}. \tag{26}$$

Plugging equation 25 and 26 back into equation 20 gives the final error bound:

**Lemma D.6** (Restatement of Lemma 3.4). *Suppose Assumptions 1–3 hold, $t > 1$, and $|\epsilon_\tau - \widehat{\epsilon}_\tau| \leq \xi_\tau$ for every $\tau \in [t-1]$. Let the NW estimators $\widehat{Q}_t$ and $\widetilde{Q}_t$ be defined as in equation 5 and 6 respectively. Then with probability at least $1 - 4T^{-2}$, for every $u \in \mathcal{C}$ and $\boldsymbol{z} \in \mathcal{Z}$, we have*

$$\left| \widetilde{Q}_t(u; \widehat{\boldsymbol{z}}) - \widehat{Q}_t(u; \boldsymbol{z}) \right| \leq \frac{1}{f_{a_t}(\boldsymbol{z})} \left( \frac{c_1}{t-1} \sum_{\tau < t} (1 + \xi_\tau)(\delta_\tau + \delta_t) + c_2 \sqrt{\frac{p^2 \log(4T)}{t a_t^p}} \right).$$

*The constant $c_1 = 2L \max\{\overline{f}_{\boldsymbol{z}}, L_{\boldsymbol{z}}\}$ and the constant $c_2$ depends on both $K$ and $\overline{f}_{\boldsymbol{z}}$.*

Note we get Lemma 3.4 by plugging in the chosen bandwidth $a_t = t^{-\frac{1}{p+2}} p^{\frac{2}{p+2}}$. Combining Lemma 3.4 with Lemma D.4 and the chosen bandwidth $a_t = c_\kappa (c_{FT}/\log(T))^{1/\omega}$, we get Lemma 3.5.

## E  OMITTED EXPERIMENT RESULTS AND DETAILS IN SECTION 5

### E.1  SYNTHETIC DATASETS

We consider the following two demand mean models:

(A) Linear demand mean: $f_*(\boldsymbol{x}_t) = \boldsymbol{\theta}_*^\top \boldsymbol{x}_t$, which is widely used in various literature.
(B) Nonlinear demand mean $f_*(\boldsymbol{x}_t) = \sin(2 \cdot \boldsymbol{\theta}_*^\top \boldsymbol{x}_t) + 2\exp(-16 \cdot (\boldsymbol{\theta}_*^\top \boldsymbol{x}_t)^2)$, which has been studied in high-dimensional quantile regression (Zhu et al., 2012) and offline robust newsvendor inventory control (Zhang et al., 2024).

For the noise models, we focus on the following three setups, where $\eta_t$ is sampled i.i.d. from standard normal distribution and truncated at $[-1, 1]$:

(1) Linear Heteroskedastic Noise (LH): $\epsilon_t = (\boldsymbol{\beta}^\top \boldsymbol{x}_t) \cdot \eta_t$, which models scenarios where noise scales linearly with features. such as customer volume or promotion intensity.
(2) Nonlinear Heteroskedastic Noise (NLH): $\epsilon_t = \sqrt{|\boldsymbol{\beta}^\top \boldsymbol{x}_t| \cdot (1 - |\boldsymbol{\beta}^\top \boldsymbol{x}_t|)} \cdot \eta_t$. This model captures situations where uncertainty is highest at intermediate levels of a contextual factor and decreases near the extremes, reflecting saturation effects or boundary constraints.
(3) Sinusoidal Heteroskedastic Noise (SH): $\epsilon_t = \sin(10 \cdot \boldsymbol{\beta}^\top \boldsymbol{x}_t) \cdot \eta_t$. This model reflects settings with periodic or seasonal variation in uncertainty, where the randomness in demand oscillates with some latent or cyclical signal.

In all settings, $\boldsymbol{\theta}_*$ and $\boldsymbol{\beta} \in \mathbb{R}^d$ are fixed, sampled from the standard multivariate Gaussian distribution and normalized to have $\ell_2$-norm bounded by 1 as well. The context vector $\boldsymbol{x}_t \in \mathbb{R}^d$ is also drawn i.i.d. from the standard multivariate Gaussian distribution and normalized to have $\|\boldsymbol{x}_t\|_2 \leq 1$. We assume that the low-dimensional feature $\boldsymbol{z}_t := \boldsymbol{\beta}^\top \boldsymbol{x}_t$ is known to the DM, but the dependence of the noise remains unknown. The intrinsic dimension in our numerical studies is thereby $p = 1$. The cost parameters in equation 1 are set to be $h = 0.05$ and $b = 0.95$. All experiments were conducted locally on a laptop with Apple M2 chip, 8-core ARM64 CPU, 8 GB memory.

First, for the linear demand mean model (A): we fix the time horizon $T = 3000$ and context dimension $d \in \{5, 10, 20\}$. We only present the comparison between OSGD in Ding et al. (2024), in which they conduct gradient descent directly over the $d$-dimensional linear coefficients, and our Algorithm 2 with ridge regression estimator in the following Figure 1. As the performance of OSGD does not significantly change with $d$ both theoretically and empirically, we only include its performance with $d = 10$ for clarity. We repeat each of the settings 20 times and plot the average cumulative regrets as well as their 95% confidence region.

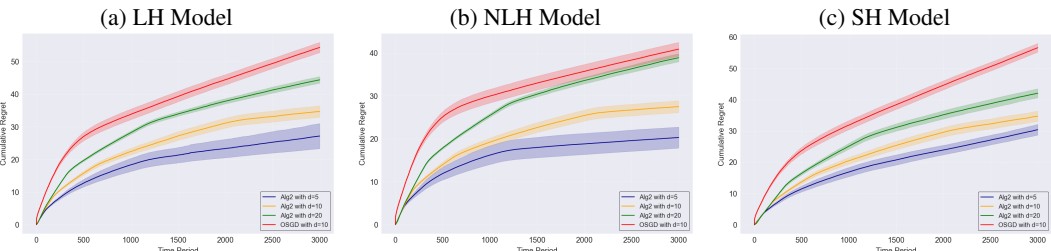

Figure 1: Regret under different noise models for linear demand mean

In the LH model, the optimal solution remains as a linear function, and hence OSGD is still theoretically optimal with $\widetilde{O}(\sqrt{T})$ regret. Yet in Figure 1(a), OSGD's performance empirically degrades under this mild heteroskedasticity. The models in Figure 1(b) and (c) are more involved and represent nonlinear and oscillatory noise structures. These are the scenarios where OSGD fails both theoretically and practically. In contrast, our algorithm consistently achieves lower regret and sublinear growth across all demand models. These numerical results highlight the significance of accounting for heteroskedasticity in practice.

For the nonlinear demand mean model (B): we fix the time horizon $T = 20000$ and context dimension $d \in \{10, 50\}$. Now we only compare the performance of OSGD and Algorithm 2 with neural network estimator. We implement a two-layer neural network with hidden dimensions 128 and 64. The

network is trained using the AdamW optimizer with learning rate $0.002$ and weight decay parameter $10^{-5}$. During training, the number of epochs and batch size are dynamically adjusted according to the current horizon: for a larger $t$, we will apply a smaller number of training epochs $J$ with a larger batch, so that we can make full use of the samples when $t$ is small and avoid a large model training time when $t$ is large. All inputs and outputs are standardized before training, and predictions are transformed back to the original scale. We repeat each of the settings 5 times (each requires a running time of around 120 minutes) and plot the average cumulative regrets as well as their 95% confidence region. The result is shown in the following figure:

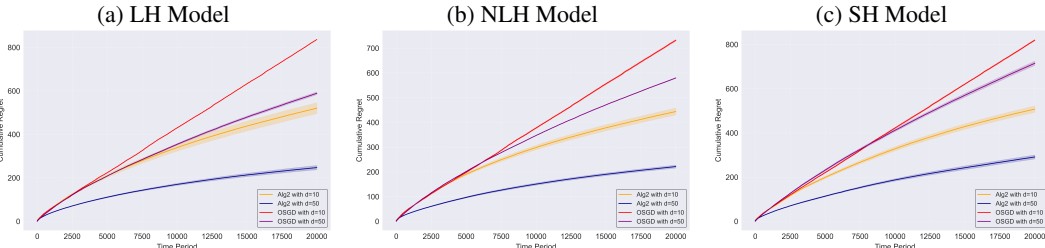

Figure 2: Regret under different noise models for nonlinear demand mean

As can be seen from Figure 2: under the highly nonlinear and oscillatory mean demand models, Algorithm 2 with the neural network estimator significantly outperforms OSGD as well. While OSGD suffers from persistent model misspecification, our approach adapts flexibly to the nonlinear structure and achieves sublinear growth of regret, which demonstrates the robustness of our algorithm against complex demand patterns. Besides, the performance of Algorithm 2 with neural networks does not necessarily degenerate as the dimension $d$ increases.

### E.2 REAL-WORLD DATASETS

We use the M5 Forecasting–Accuracy dataset from Kaggle, which is widely used for evaluating forecasting and inventory management algorithms. The dataset contains daily sales records of more than 30,000 Walmart products across three U.S. states (California, Texas, and Wisconsin), covering a time horizon of 1,941 days. In addition to sales quantities, it provides rich feature information such as item attributes (category, department), store/location information, calendar variables (events, holidays), and price data (including temporary promotions). See Howard et al. (2020) for a detailed illustration of feature information.

Among all the products, we select 40 items that have the most nonzero selling periods. Specifically, for all these items, they have selling records of more than 1,900 days. The items are all under the food category, and the average sales count is 35.2 units per day. Before observing the demand realization, the DM can observe a $d = 24$ dimensional feature vector that helps her make an inventory order decision.

To see that the context-aware noise is indeed an issue in the real-world dataset, we first present a box plot on how the demand mean and variance varies with different values of feature. We consider two features: (1) "Is Weekend", where 1 denotes the weekend and 0 denotes the weekday; (2) "Sell Price", where we have 3 different prices in total.

We can see from Figure 3 that: (1) for Is Weekend, both the demand mean and variance of demand differ significantly between weekdays and weekends, indicating that sales behavior changes systematically across these contexts; (2) for different Sell Price, while the mean demand across the three price does not differ substantially, at the highest price (1.68), the variance of demand is significantly larger. These results clearly indicate that the variance of demand depends on certain contextual features. As we argued earlier, models with context-independent noise cannot capture such heteroskedastic patterns. This highlights the practical importance of studying our proposed formulation with context-aware noise.

For the details of implementing Algorithm 2 in this real-world dataset: Note that we do not know exactly the true low-dimensional feature $z_t$ (e.g. as a function of the context $x_t$). Instead, we will use the estimated demand mean $\widehat{z}_t := \widehat{f}_t(x_t)$ at each period $t$ as a hopefully good approximation of

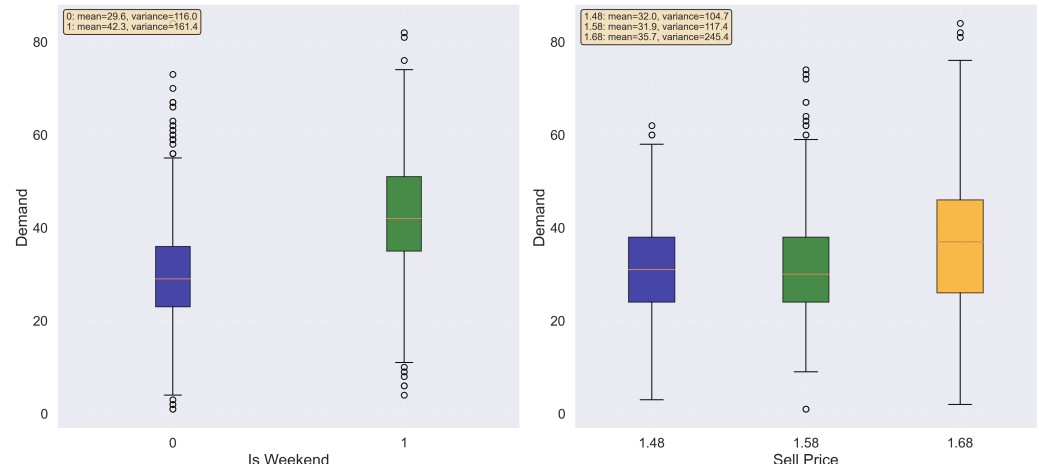

Figure 3: Context-aware heteroskedasticity in real-world dataset

$z_t$, with $p = 1$. The details of ridge regression and neural networks are the same as in the synthetic datasets in Appendix E.1.

