# OpenReview forum: "Perishable Online Inventory Control with Context-Aware Demand Distributions"
_ICLR.cc/2026/Conference — Submitted to ICLR 2026_

### Official Review · Reviewer_Ko2b · 2025-10-14

**Soundness:** 3
**Presentation:** 3
**Contribution:** 2
**Rating:** 4
**Confidence:** 2

**Summary:**

This paper studies online contextual inventory control with perishable goods when both the demand mean and the residual noise distribution depend on observable features. The key departure from prior work is to allow heteroskedastic, context-dependent noise via a p-dimensional feature vector z_t (possibly a transform or subset of x_t), which makes the optimal order quantity no longer linear in x_t and breaks the usual OSGD approach that relies on linearity under i.i.d. noise. The authors propose a plug-in, estimation-to-decision algorithm: estimate the conditional mean demand f*(x_t) with either ridge (linear) or over-parameterized neural networks (nonlinear), then estimate the conditional CDF of the noise Q(·; z) using a Nadaraya–Watson estimator built from estimated residuals and (possibly noisy) features, and finally choose the inventory c_t by maximizing the corresponding plug-in newsvendor objective $,\tilde{\ell}_t(c)=h\int_0^c \hat{Q}_t(y-\hat{D}_t),dy + b\int_c^M (1-\hat{Q}_t(y-\hat{D}_t)),dy$. Theoretically, for linear f*, they prove a regret upper bound $\tilde{O}(\sqrt{dT} + T^{\frac{p+1}{p+2}})$ and a matching minimax lower bound $\Omega(\sqrt{dT}+T^{\frac{p+1}{p+2}})$, identifying p as an intrinsic dimension of the noise’s context dependence. For nonlinear f*, they train a sufficiently wide and deep network (NTK regime) and obtain regret $\tilde{O}(\sqrt{\bar{d},T}+T^{\frac{p+1}{p+2}})$ where $\bar{d}$ depends on the effective dimension of the NTK kernel and the function norm of $f^*$. Under an additional Fourier-decay condition on the feature/noise distributions, the nonparametric term improves to $\tilde{O}(\sqrt{pT})$. Experiments on synthetic settings and the M5 dataset show that the proposed method outperforms an OSGD baseline when noise is heteroskedastic or the mean is nonlinear.

**Strengths:**

(1) The efforts to resolve nonlinearity (i.e. involvement of neural networks) in contextual inventory control problems: Regret bound in the NTK/over-parameterized regime, tying complexity to an effective dimension and function norm.

(2) The rigorous proofs of matching upper and lower regret bounds $\tilde{O}(\sqrt{dT}+T^{\frac{p+1}{p+2}})$, which closes the information-theoretic gap.

(3) Empirical results support the claim that OSGD breaks under heteroskedastic noise or nonlinear means.

**Weaknesses:**

The authors are encouraged to disagree with my points. Convincing rebuttals will end up with a higher evaluation.

(1) Novelty of the nonlinear part is mostly in assembling well-known tools. The paper’s nonlinear extension is interesting for this domain, but methodologically it relies on standard statistical covering arguments and NTK-based local linearization; the hypothesis space here is not especially exotic, and the complexity control mostly follows known templates. The value is in adapting these to perishable inventory with heteroskedasticity, rather than a fundamentally new learning technique.

(2) Assumption stack can be heavy for the nonlinear case. The NTK analysis requires very wide nets, carefully chosen stepsizes, and a non-singular NTK over the random contexts. The required over-parameterization and effective-dimension conditions may be far from practice, even if small networks work empirically; the theory may overstate what is needed in real deployments.

**Questions:**

1, On the role and selection of $p$: In practice, how should one choose or validate the feature map $z_t$ and its dimension $p$? For instance, when using $\hat{z}_t := \hat{f}_t(x_t) (p=1)$, are there diagnostics that indicate when this is insufficient, and would you recommend simple expansions like $z_t = [\hat{f}_t(x_t), x_t,j]$ for a few $j$?

2, Sensitivity to local density via $\hat{f}_a(z)$. The pointwise error bounds scale with $1/\hat{f}_a(z)$, and you control the effect through a probability argument. Empirically, did you observe instability when $z_t$ visits low-density regions (e.g., early rounds or rare contexts)? Do you implement any floor on $\hat{f}_a(z)$ or adaptive bandwidths to stabilize estimates?

3, Disclosure of LLM usage: As is required to have an explicit statement on the LLM usage of this work, do you have a paragraph on that?

---

### Official Review · Reviewer_FqGD · 2025-10-24

**Soundness:** 3
**Presentation:** 3
**Contribution:** 3
**Rating:** 4
**Confidence:** 4

**Summary:**

This paper studies online contextual inventory control where both the expected demand and the noise distribution depend on observable features, modeling context-aware heteroskedastic demand. The authors provide an online-learning algorithm combining: a mean-estimation oracle (either ridge regression for linear demand or over-parameterized neural networks for nonlinear demand), and a kernel regression estimator for the noise CDF with measurement and observation errors. They also show the regret upper bound of the algorithm, together with an almost-matching lower bound, which can be reduced to classical results with stronger noise assumptions.

**Strengths:**

1. The contextual heteroskedastic noise setting is original and practically motivated. Modeling context-dependent uncertainty is realistic.
2. The theoretical results are complete. The minimax-optimal characterization of regret is interesting. The integration of nonparametric CDF estimation and learning-while-optimizing also appears to be novel.
3. The paper is well written and easy to follow.

**Weaknesses:**

### Scope and Audience ###
I am slightly concerned about whether this paper has a sufficiently broad audience at ICLR. While it falls under the umbrella of online learning, the problem formulation and motivation are quite domain-specific to inventory management. I will not push strongly in this direction, but I would leave it to the senior reviewers and ACs to decide whether the paper fits the conference scope.

### Practical Relevance of the Setting ###
Although there has been some work on applying online learning to inventory decisions, it remains debatable whether online learning is the right paradigm for such problems. In practice, inventory decisions are not made frequently, typically on a weekly or even monthly basis. In the numerical experiment, Figure (c) shows that both proposed algorithms begin to outperform OSGD only after about 100 days, which is quite long in a practical context. In real-world implementations, such algorithms would likely be shoot down before they exhibit a performance advantage.

### Using of "Perishable goods" ###
The paper describes the setting as involving perishable goods, but the model essentially corresponds to a classical single-period (newsvendor-type) problem where unsold inventory is discarded at the end of each period. In more recent literature, perishable goods are referred to those that will stay for (only) a couple of periods, which make the state space much larger than the classical single-period problem. I would suggest the authors to reconsider the terminology used.


### Technical Questions ###
I have several technical questions listed below.

**Questions:**

I have some technical questions:

1. I am not very clear about how to make the constructed Q_t(y) zero-mean? In the section A.1., the authors mentioned that "through tuning the first component of $\theta_*$ and $x_t$." but the $x_t$ and $\theta_*$ have been defined very well. Could the authors provide more specific guidelines on this?
2. As everything is bounded (please correct me if I am wrong), I can always ignore the heteroskedastic noise and just think it is generated from a sub-Gaussian distribution (maybe with a large variance proxy). By doing this, I think I can always get a $\sqrt{T}$ bound. Am I missing something?

---

### Official Review · Reviewer_g2MK · 2025-10-31

**Soundness:** 3
**Presentation:** 3
**Contribution:** 3
**Rating:** 6
**Confidence:** 4

**Summary:**

This paper addresses online contextual inventory control for perishable goods under a more realistic setting where both the expected demand and noise distribution depend on observable features. The authors propose an algorithm combining demand mean estimation (via ridge regression or neural networks) with kernel regression for CDF estimation. They establish a minimax regret lower bound of $\Omega(\sqrt{dT} + T^{(p+1)/(p+2)})$ and prove their algorithm achieves near-optimal regret $\tilde{O}(\sqrt{dT} + T^{(p+1)/(p+2)})$ for linear demand and $\tilde{O}(\sqrt{\alpha T} + T^{(p+1)/(p+2)})$ for nonlinear demand. Under additional regularity conditions, the exponential term improves to $p\sqrt{T}$.

I am very familiar with both the inventory control literature and contextual bandit/online learning theory. I have carefully verified the mathematical arguments and compared with related work in both areas. The technical content is sound, and my assessment focuses primarily on presentation, practical applicability, and positioning relative to existing literature.
References
[1] Ding, J., Huh, W. T., & Rong, Y. (2024). Feature-based inventory control with censored demand. Manufacturing & Service Operations Management, 26(3), 1157-1172.
[2] Bu, J., Simchi-Levi, D., & Xu, Y. (2020, November). Online pricing with offline data: Phase transition and inverse square law. In International Conference on Machine Learning (pp. 1202-1210). PMLR.
[3] Ao, R., Jiang, J., & Simchi-Levi, D. (2025). Learning to Price with Resource Constraints: From Full Information to Machine-Learned Prices. arXiv preprint arXiv:2501.14155.

**Strengths:**

⦁	Clear and compelling motivation: The introduction effectively argues why standard homoskedastic assumptions are unrealistic, with concrete examples (umbrella sales, binomial demand) illustrating when heteroskedasticity naturally arises.
⦁	Strong theoretical completeness: The paper provides both minimax lower bounds and matching upper bounds (up to logarithmic factors), offering a complete characterization of the problem's fundamental difficulty. The analysis elegantly decomposes the problem into mean estimation and non-parametric regression components.
⦁	Novel algorithmic approach: The core idea of using hierarchical elimination to ensure conditional independence while combining contextual bandits techniques with kernel regression is creative and technically sound.
⦁	Extension to nonlinear demand: The treatment of nonlinear demand via neural networks and NTK analysis represents a significant advance, as prior online inventory control literature has focused primarily on linear models.
⦁	Empirical validation: Experiments on both synthetic and real-world (M5 dataset) data demonstrate practical superiority over OSGD baselines, particularly under heteroskedastic conditions.

**Weaknesses:**

Algorithmic clarity and intuition:
⦁	The three if-else conditions in Algorithm 2 lack detailed intuitive explanations. More high-level discussion of the scenarios corresponding to each condition and the motivations for the associated actions would significantly improve accessibility.
⦁	The connection to hierarchical elimination from the bandit literature (Auer 2002, Chu et al. 2011) could be explained more explicitly to clarify what is borrowed versus what is novel for this specific inventory control setting.
Analysis of OSGD failure modes:
⦁	While the paper claims OSGD fails under context-dependent noise, more concrete analysis of how and why OSGD deteriorates would strengthen the motivation. Does it suffer constant regret, or merely worse rate dependence? Specific examples demonstrating failure would be valuable.
Strong assumptions:
⦁	Known feature mapping $\psi(x_t)$: The assumption that the practitioner knows the correct feature mapping determining noise dependence is quite strong. The paper acknowledges this briefly but does not adequately address the practical challenge of discovering this mapping. How sensitive is performance to misspecification of $\psi$?
⦁	Over-parameterization for neural networks: The width requirements ($w = \tilde{\Omega}(\text{poly}(T, K, \lambda^{-1}))$) for the NTK analysis are extremely large and impractical. While the experiments show small networks work well, the theory-practice gap deserves more discussion.
Limited comparison with related work:
⦁	The paper would benefit from more detailed comparison with the contextual bandit literature (Auer 2002, Chu et al. 2011, Han et al. 2025, Wen et al. 2025) to make the unique difficulties of the inventory control setting clearer.
⦁	Discussion of how offline data or machine learning oracles (references [2], [3] in the review) could improve the minimax results would provide valuable practical direction.
Experimental limitations:
⦁	Real-world experiments focus only on food items from the M5 dataset. Evaluation on other categories or datasets would strengthen generalizability claims.
⦁	The synthetic experiments use relatively simple noise structures (linear, binomial, sinusoidal). More complex heteroskedastic patterns could better stress-test the algorithm.
Missing practical guidance:
⦁	How should practitioners determine the feature mapping $\psi$ in practice? The umbrella example suggests domain knowledge, but more systematic approaches would be valuable.
⦁	Guidance on bandwidth selection and hyperparameter tuning for the kernel regression component is limited.

**Questions:**

See above.

---

### Official Review · Reviewer_LTuu · 2025-11-01

**Soundness:** 2
**Presentation:** 2
**Contribution:** 2
**Rating:** 6
**Confidence:** 2

**Summary:**

This paper studies an online demand learning problem with perishable items where the decision maker (DM) observes a context $x_t$, selects an inventory level $c_t$, and then observes the full demand $D_t = f_\star(x_t) + \epsilon_t$.
Unlike previous works, this paper assumes that the noise follows a context(feature)-dependent conditional distribution
$\epsilon_t \sim Q(\cdot; z_t)$ where feature $z_t$ is a transformation of the context $x_t$. This setting departs from previous works that assume i.i.d. or homoskedastic noise. They establish the minimax regret bound for linear expected damand model with Heteroskedastic noise. Moreover, under a mild regularity condition (a Fourier-smoothness assumption), the regret bound improves to \tilde{O}(\sqrt{\alpha T} + p\sqrt{T}). Additionally, they also derived the regret upper bound for non-linear model.

**Strengths:**

1. Feature-dependent noise modeling.
- The key novelty lies in modeling heteroskedastic, feature-dependent noise rather than assuming i.i.d. noise. This is a realistic modeling, and the paper clearly motivates how noise variability depends on contextual features in practical demand systems.

2. Tight regret bound for online demand learning with context-dependent noise.
- The regret bound clearly decomposes into a parametric term $\tilde{O}(\sqrt{\alpha T})$ and a nonparametric term $\tilde{O}(T^{(p+1)/(p+2)})$. In particular, for the linear case, this bound matches the minimax regret lower bound.

**Weaknesses:**

The theoretical results appear technically sound, and no major issues were found.

However, several aspects of the presentation could be improved.

1. Justifying that Assumption 1 (Lipschitz CDF) is a standard and mild regularity assumption (as commonly used in nonparametric statistics) would be helpful.
2. A comparison with Ding et al. (2024) would strengthen the discussion. Ding et al. (2024) assumes i.i.d. noise and achieves $O(d\log T)$ regret, while this paper obtains $O(\sqrt{dT})$ under feature-dependent heteroskedastic noise.
Explaining this difference—e.g., that $\sqrt{dT}$ arises from jointly estimating $\theta^*$ and the noise structure (as noted in Appendix A.1)—would improve interpretability.
3. The experimental section is somewhat limited. For example, showing that the real-world data exhibit heteroskedastic noise is needed. The explanation for Figure (a) also needs to be more detailed.

**Questions:**

1. Comparison to Ding et al. (2024)
- Does the factor $\sqrt{T}$ arise from estimating true heteroskedastic noise? Why does your result improve over Ding et al. (2024) by a factor of \sqrt{d}? Can you compare your results with Ding et al. (2024)?

2. Numerical results
- Why does the NN estimator sometimes outperform the Ridge regression estimator, while in other cases the Ridge estimator performs better, as observed in Figures (b) and (c)?

---

### Official Review · Reviewer_1c9Q · 2025-11-01

**Soundness:** 2
**Presentation:** 1
**Contribution:** 1
**Rating:** 2
**Confidence:** 4

**Summary:**

This paper considers online perishable inventory problem with contextual information. Because of perishability, there is no inventory carryover, the problem setup is more akin to a newsvendor inventory problem under bandit setting. The authors consider three types of demand estimation and derive regret bound correspondingly: linear demand, neural network parametrized demand, kernel density function is demand.

**Strengths:**

- Online inventory control is an important application domain, and the authors also conduct experiments on real data.

**Weaknesses:**

- The problem setting regarding the demand information is very strong, i.e., full demand information without demand censoring. Given that censored feedback is a vital issue in inventory control, the setup is too simple and not realistic. Importantly, a key prior work considers censored demand feedback, and this is also considered in the non-contextual case.
- The overall organization lacks clarity. There is not an unified framework on the three types of demand functions considered; instead, it is a collection of existing estimation accuracy results of three offline settings, with straightforward adaption to the online setting.

[1] Ding J, Huh WT, Rong Y. Feature-based inventory control with censored demand. Manufacturing & Service Operations Management. 2024 May;26(3):1157-72.

[2] Zhang W, Li C, Qin H, Xu Y, Zhu R. Thompson Sampling for Repeated Newsvendor. arXiv preprint arXiv:2502.09900. 2025 Feb 14.

**Questions:**

- Given the current setting with full information and no demand censoring, how is it qualitatively different from contextual continuum arm bandit problems? Are there any particular challenges or methodological advances in this work?
- Can the authors articulate more on the motivation of context-dependent noise. Given that mean demand is contextual dependent and the simple form of the newsvendor loss, what is the necessity of this particular modeling? It's also helpful to give some examples of the noise distribution.
- Line 8 in Algorithm 2 appears to be a typo. Does this affect the proof argument?
- The numerical study lacks natural baseline comparisons. How would a generic online quantile regression approach work? Some more implementation details such as neural network size, kernel bandwidth would also help with reproducibility.

---

### Meta-Review · Area_Chair_omD9 · 2026-01-04

**Summary:**

The paper investigates the online contextual inventory control problem specifically for perishable goods, where unsold inventory is discarded at the end of each period. The core novelty lies in modeling heteroskedastic, context-dependent noise where both the expected demand and the residual noise distribution depend on observable features.

The main concerns that drove the decisions for rejection of the paper includes:

1. The paper assume we see "full demand," but in real life we have censored demand (we don't know how many people wanted to buy if we run out of stock).
2.  Some reviewers claim that the method is just "assembling well-known tools" and not super new in how it learn
3.  One reviewer noticed that the algorithm takes 100 days to be better than OSGD, which is too slow for real world scenarios.
4. There is a suggestion that "perishable" is used loosely, as the model is a "classical single-period (newsvendor-type) problem" rather than a multi-period state-space problem

**Reviewer Concerns:**

NA - the authors has not submitted the rebuttal

**Reviewer Scores:**

NA - the authors have not submitted the rebuttal.

The overall scores have indicated a borderline recommendation, the paper is interesting but would need some work to be considered for ICLR.

---

### Decision · Program_Chairs · 2026-01-26

Reject